# MPI depletion enhances O-GlcNAcylation of p53 and suppresses the Warburg effect

Nataly Shtraizent[1,2†], Charles DeRossi[1,2†], Shikha Nayar[1,2], Ravi Sachidanandam[3], Liora S Katz[4], Adam Prince[1], Anna P Koh[5], Adam Vincek[6], Yoav Hadas[6], Yujin Hoshida[5], Donald K Scott[4], Efrat Eliyahu[6], Hudson H Freeze[7], Kirsten C Sadler[8], Jaime Chu[1,2*]

[1]Department of Pediatrics, Icahn School of Medicine at Mount Sinai, New York, United States; [2]The Mindich Child Health and Development Institute, Icahn School of Medicine at Mount Sinai, New York, United States; [3]Department of Oncological Sciences, Icahn School of Medicine at Mount Sinai, New York, United States; [4]Department of Medicine, Division of Endocrinology, Diabetes and Bone Disease, Icahn School of Medicine at Mount Sinai, New York, United States; [5]Department of Medicine, Division of Liver Diseases, Icahn School of Medicine at Mount Sinai, New York, United States; [6]Department of Genetics and Genomic Sciences, Icahn Institute for Genomics and Multiscale Biology, Icahn School of Medicine at Mount Sinai, New York, United States; [7]Sanford Children's Health Research Center, Sanford Burnham Prebys Medical Discovery Institute, La Jolla, United States; [8]Biology Program, New York University Abu Dhabi, Abu Dhabi, United Arab Emirates

*For correspondence: jaime.chu@mssm.edu

[†]These authors contributed equally to this work

Competing interests: The authors declare that no competing interests exist.

**Abstract** Rapid cellular proliferation in early development and cancer depends on glucose metabolism to fuel macromolecule biosynthesis. Metabolic enzymes are presumed regulators of this glycolysis-driven metabolic program, known as the Warburg effect; however, few have been identified. We uncover a previously unappreciated role for Mannose phosphate isomerase (MPI) as a metabolic enzyme required to maintain Warburg metabolism in zebrafish embryos and in both primary and malignant mammalian cells. The functional consequences of MPI loss are striking: glycolysis is blocked and cells die. These phenotypes are caused by induction of p53 and accumulation of the glycolytic intermediate fructose 6-phosphate, leading to engagement of the hexosamine biosynthetic pathway (HBP), increased O-GlcNAcylation, and p53 stabilization. Inhibiting the HBP through genetic and chemical methods reverses p53 stabilization and rescues the Mpi-deficient phenotype. This work provides mechanistic evidence by which MPI loss induces p53, and identifies MPI as a novel regulator of p53 and Warburg metabolism.

## Introduction

The importance of glucose metabolism to cellular proliferation was uncovered by the pioneering work of Otto Warburg who showed that cancer cells undergo an energetic shift from oxidative phosphorylation to increased aerobic glycolysis (*Warburg et al., 1924*). This metabolic reprogramming, termed the Warburg effect, is required for the rapid proliferation of cancer cells. This strategy has also been shown to be utilized by rapidly dividing cells during development (*Krisher and Prather, 2012*; *Ma et al., 2010*): retinal progenitor cells (*Agathocleous et al., 2012*), proliferating lymphocytes (*Fox et al., 2005*; *Palmer et al., 2015*) and thymocytes (*Brand and Hermfisse, 1997*), and embryonic stem cells (*Kondoh et al., 2007*; *Kim et al., 2014*), all of which rely on glycolysis as the primary mechanism of energy generation. Thus, as with many cancer-promoting pathways, the

Warburg effect appears to be an embryonic program co-opted by cancer cells. However, the physiological and molecular factors that regulate the Warburg effect in both cancer and embryonic cells remain poorly understood.

Signaling and tumor suppressor pathways, including the AKT/PI3K and p53 pathways, have been shown in some systems to regulate glycolysis (*Bensaad et al., 2006*; *Elstrom et al., 2004*; *Kruiswijk et al., 2015*). However, there is only a limited understanding of the metabolic control points and key players involved in the complex relationship between metabolism and cell proliferation. Despite intensive work on the Warburg effect over the past 90 years, the mechanisms that promote this dependency on glycolysis for cell survival are poorly understood. Clarifying the complex regulatory mechanisms of cells that employ the Warburg effect could reveal means to manipulate this fuel-generating mechanism of cancer cells. Metabolic reprogramming in cancer cells occurs by inducing the expression and activity of glycolytic enzymes. The paradigm is shifting with the discovery that metabolic enzymes are not merely secondary bystanders, but function as central oncogenic players. For instance, glycolytic enzymes such as PFK and LDHA have been shown to be essential for driving tumor formation and growth (*Fantin et al., 2006*; *Le et al., 2010*; *Yalcin et al., 2009*), and tumor suppressor genes such as p53 are emerging as primary regulators of energy production (*Bensaad et al., 2006*; *Cheung and Vousden, 2010*; *Hu et al., 2010*).

Mannose phosphate isomerase (MPI) catalyzes the interconversion of mannose 6-phosphate (Man6P) and fructose 6-phosphate (Fru6P)(*Gracy and Noltmann, 1968*), effectively bridging N-glycosylation pathways with energy metabolism pathways, such as glycolysis and the hexosamine biosynthetic pathway (HBP; *Figure 1*). MPI is highly conserved from yeast to humans (*Schollen et al., 2000*). No humans with complete absence of activity have been identified, suggesting that total loss

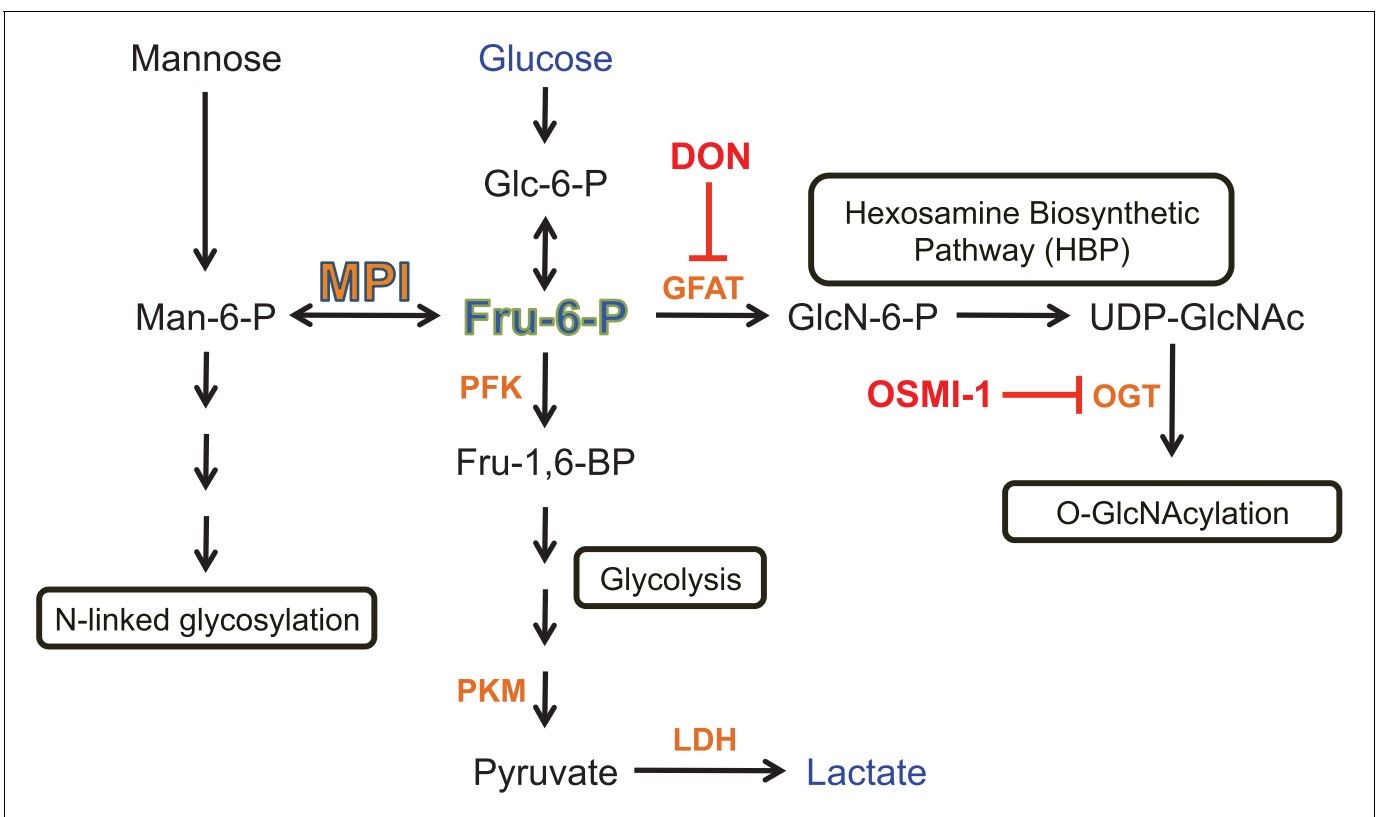

**Figure 1.** Schematic illustration of involvement of MPI and Fru6P in metabolic pathways. Key points described in the scheme are: MPI interconverts Man6P and Fru6P, and Man6P is a precursor of N-glycosylation. Fru6P is a substrate of PFK in glycolysis or a substrate of GFAT in HBP. PFK is a central rate-limiting enzyme in glycolysis; LDH catalyzes formation of lactate from pyruvate. DON is an inhibitor of GFAT, and as such an inhibitor of HBP. OGT is the primary enzyme that catalyzes the addition of O-GlcNAC to proteins. Metabolites measured in this study are shown in blue (glucose and lactate) and green (Fru6P). Chemical inhibitors used in this study are shown in red. Relevant enzymes are shown in orange.

of MPI is incompatible with survival. This conclusion is consistent with the findings that knockout of *Mpi* in mice is embryonic lethal by E11.5 (*DeRossi et al., 2006*), and significant decrease of Mpi activity is embryonic lethal in zebrafish (*Chu et al., 2013*). Partial loss of MPI function in humans leads to a congenital disorder of glycosylation (CDG; MPI-CDG) characterized by hepatic fibrosis, biliary malformations, protein-losing enteropathy, and coagulopathy (*de Koning et al., 1998*; *Jaeken et al., 1998*; *Niehues et al., 1998*). However, the clinical presentation of MPI-CDG is inexplicably unique with its absence of neurologic or musculoskeletal disease when compared to every other 40+ types of CDG affecting N-linked glycosylation (*Sparks and Krasnewich, 2014*; *de Lonlay et al., 2001*). This stark clinical discrepancy, along with studies with knockout *Mpi* mouse models where N-glycans remain largely intact (*Higashidani et al., 2009*; *DeRossi et al., 2006*), raise the possibility that MPI may play roles in other cellular processes. Little is known about the function of MPI in cancer, but a recent study showing that MPI depletion in brain cancer cells enhances radiation-induced cell death (*Cazet et al., 2014*) supports our findings that Mpi loss in zebrafish embryos promotes cell death, and indicates that MPI provides a strong survival advantage in both embryos and cancer cells.

In this study, we describe a previously unappreciated function of MPI as a glycolytic regulator, separate from its effects on protein N-glycosylation, and its important contribution to the metabolic regulation driving cellular proliferation in embryonic development and cancer. We report that MPI loss suppresses glycolysis and stabilizes p53 leading to increased apoptosis. We provide mechanistic evidence showing that MPI loss leads to the accumulation of Fru6P and increased O-GlcNAcylation of proteins, notably p53, which results in its stabilization. Inhibition of the hexosamine biosynthetic pathway (HBP) or O-GlcNAcylation destabilizes p53 and rescues the Mpi-deficient phenotype and apoptosis. This demonstrates that MPI promotes the Warburg effect and cell survival, whereby loss of MPI can suppress glycolysis and activate O-GlcNAcylation and p53 through regulation of Fru6P accumulation and HBP activity. These mechanistic insights that may not only account for the clinical divergence of MPI-CDG from other CDG subtypes but also uncovers this novel pathway as a metabolic strategy that is conserved across vertebrates, common to both embryonic development and cancer, and broadens our understanding of the regulation of Warburg metabolism.

## Results

### Mpi loss causes cell death in zebrafish embryos via activation of p53

The stark discrepancy in clinical presentation of MPI-CDG patients with unique absence of debilitating neuromuscular symptoms found with every other type of CDG prompted us to investigate whether MPI plays a role in a cellular process distinct from protein N-glycosylation. To identify the molecular pathways that are activated as a result of Mpi depletion, we performed unbiased gene expression profiling using RNA-seq analysis on *mpi* morphant (MO) zebrafish embryos at 24 hr post fertilization (hpf), compared with embryos injected with a standard control (std) MO. Effectiveness of the morpholino targeting *mpi* was confirmed by the decrease in Mpi enzymatic activity to 27% of controls (*Figure 2—figure supplement 1A*), which we have previously shown to have no effect on another enzyme in the mannose metabolism pathway, phosphomannomutase 2 (Pmm2)(*Chu et al., 2013*), and the morphant phenotype could be rescued by either *mpi* mRNA coinjection or by mannose supplementation (*Chu et al., 2013*), the latter being the cornerstone of treatment for MPI-CDG patients (*Niehues et al., 1998*).

RNA-seq analysis of Mpi-depleted zebrafish embryos revealed only modest changes in genes involved in N-glycosylation (*Figure 2—figure supplement 1B*; *Figure 2—source data 1*). Instead, expression of *tp53 (p53)* and its downstream targets associated with cell cycle regulation and apoptosis were among the highest up-regulated genes in *mpi* morphants (*p53* fold change (FC) log2 value of 2.8; *Figure 2A* and *Figure 2—source data 2*). Candidate genes selected from the panel (*tp53, baxa, p21, casp8 and mdm2*) were validated on at least three additional biological samples by qPCR (*Figure 2—figure supplement 2A*: *p53*: N = 10, FC = 1.8, p=0.0002; *p21*: N = 8, FC = 13.9, p=0.0006; *mdm2*: N = 4, FC = 8.2, p=0.002; *baxa*: N = 4, FC = 2.3, p=0.001; *casp8*: N = 3, FC = 17.3, p=0.009). p53 protein levels were also elevated in response to loss of Mpi. In 12 independent *mpi* MO samples at 24 hpf, p53 protein was significantly increased 3.2-fold (N = 12, p<0.0001), as compared to embryos injected with std MO, where essentially no p53 was detected (*Figure 2—*

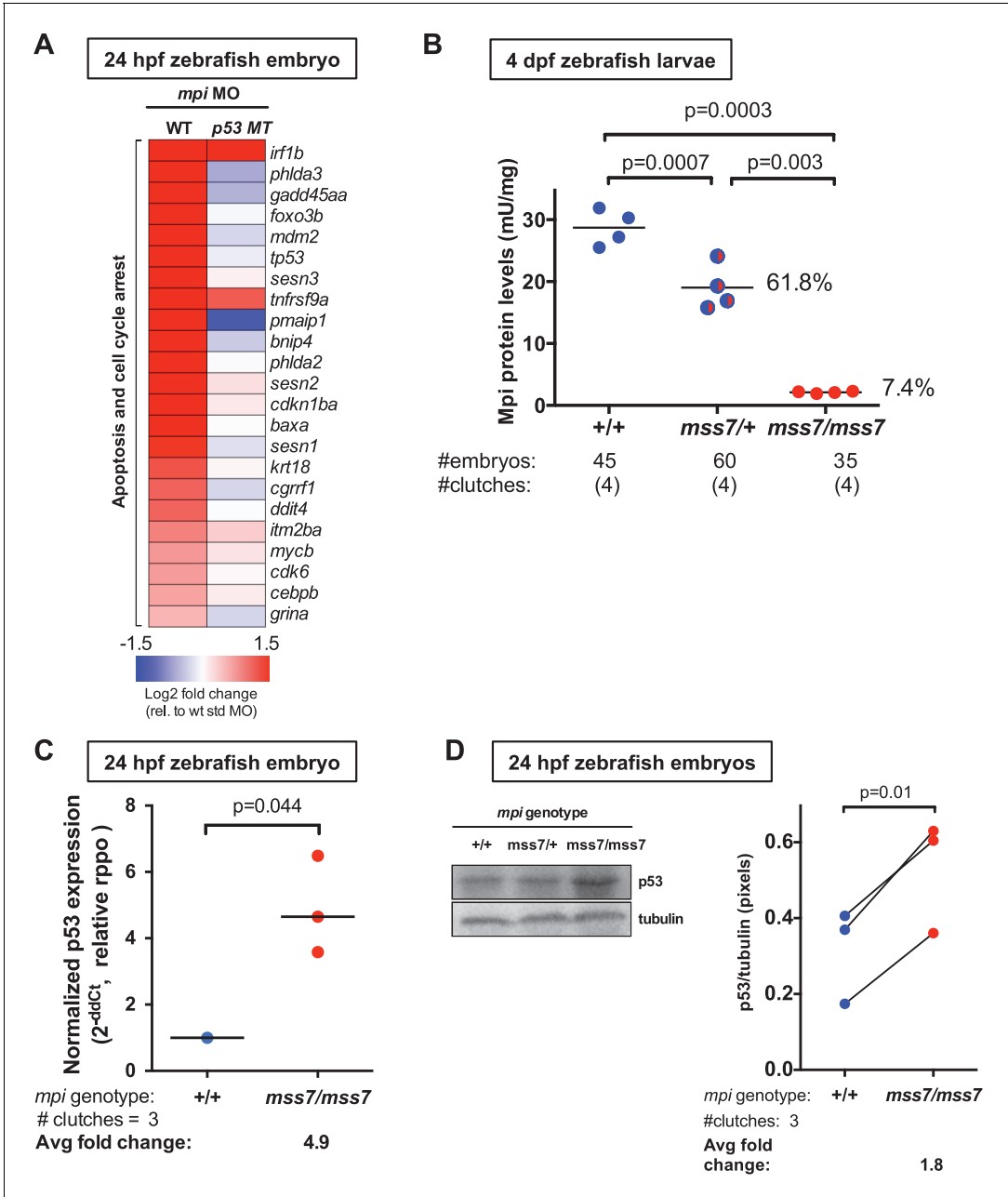

**Figure 2.** Mpi loss causes cell death in zebrafish embryos via activation of p53. (**A**) Whole transcriptome analysis of Mpi-depleted zebrafish embryos using RNA-seq showed increased *p53* and apoptosis-related genes. Standardized fold change (FC) in *mpi* morphants in WT and *p53* MT was calculated relative to standard control morpholino (std MO)-injected embryos. The color scheme represents gene expression changes in a log2 scale, in the range of −1.5 (blue, decreased) to 1.5 (red, increased). (**B**) Residual Mpi activity is significantly decreased at 4 dpf in $mpi^{mss7}$ MT larvae, compared to $mpi^{+/+}$ siblings. (**C**) *p53* mRNA levels are upregulated as analyzed through qPCR analysis in 24 hpf $mpi^{mss7/mss7}$ embryos, compared to $mpi^{+/+}$ siblings. (**D**) p53 protein levels significantly increased at 24 hpf in $mpi^{mss7/mss7}$ embryos assessed by western blot, as compared to $mpi^{+/+}$ siblings. Western blots quantified using densitometry analysis (ImageJ). p-Value based on two-tailed paired one sample t-test.

The following source data and figure supplements are available for figure 2:

**Source data 1.** Changes in expression of the genes involved in the N-glycosylation pathway.

**Source data 2.** Changes in expression of the genes involved in apoptosis and cell cycle arrest.

**Figure supplement 1.** Mpi-depleted zebrafish embryos show few changes in N-glycosylation.

*Figure 2 continued on next page*

*Figure 2 continued*

**Figure supplement 2.** p53 and its targets are increased in Mpi-depleted zebrafish embryos.

**Figure supplement 3.** *mpi* ^mss7^ MT are embryonic lethal.

*figure supplement 2B*). Furthermore, p53 protein remained elevated in *mpi* MO embryos through 4 dpf compared to std MO (N = 10, p=0.002; *Figure 2—figure supplement 2C*), indicating that p53 stabilization in Mpi-depleted embryos is maintained throughout embryonic development.

To rule out a morpholino-specific effect, these results were recapitulated in an *mpi* mutant line generated using transcription activator-like effector nuclease (TALEN) gene editing to target exon 2 (E2) of the zebrafish *mpi* genomic sequence (ZFIN: *mpi*^mss7^; *Figure 2—figure supplement 3A–B*). The resulting indel mutation is predicted to produce a p.Gln9_Ala12delinsProPro substitution at the amino acid level and is in a region that is moderately conserved from yeast to humans (*Figure 2—figure supplement 3C*). Notably, *mpi*^mss7/mss7^ homozygous and *mpi*^mss7/+^ heterozygous 5 dpf larvae had reduced Mpi enzyme activity to 7.4% (N = 4 independent clutches, p<0.0001) and 61.8% (N = 4, p=0.015) of controls, respectively (*Figure 2B*). Survival analysis of offspring from *mpi*^mss7/+^ heterozygous incrosses revealed reduced clutch size by 13 dpf, compared to control WT incrosses (50% survival vs. 77.3% survival, respectively), indicating increased lethality associated with *mpi* mutation (*Figure 2—figure supplement 3D*). Genotyping sample offspring from *mpi*^mss7/+^ incrosses confirmed a reduced occurrence of *mpi*^mss7/mss7^ homozygotes (*Figure 2—figure supplement 3E*), a result consistent with previous reports in mice (*Sharma et al., 2014*).

Importantly, both *p53* mRNA (N = 3, FC = 4.9, p=0.044; *Figure 2C*) and p53 protein levels at 24 hpf were elevated in *mpi*^mss7/mss7^ embryos (N = 4 clutches, FC = 1.8, p=0.01; *Figure 2D*) compared to *mpi*^+/+^ siblings. This demonstrates that p53 induction is a true response to Mpi depletion in early embryos, and not due to off-target effects from MO-related induction of p53 (*Robu et al., 2007*; *Kok et al., 2015*). Together, these data reveal Mpi as necessary for embryogenesis and cell survival, and that loss of Mpi induces p53.

Given the similarity of p53 induction between *mpi* morphant and mutants, we carried out further studies with both knockdown models to permit a necessary titration of Mpi levels, and allow for more detailed dissection of the relationship with p53 in response to acute Mpi depletion (*mpi* MO) and stable Mpi attenuation (*mpi*^mss7^ mutation). We next investigated the functional relationship between p53 and Mpi by co-injection of the morpholino targeting *mpi* with one specific for *tp53* (*p53* MO), or, alternatively, into *p53*-deficient mutant zebrafish (*p53*^e7/e7^, herein referred to as *p53* MT) (*Berghmans et al., 2005*). We found that blocking p53 substantially rescued all morphological phenotypes in *mpi* MO embryos (*Figure 3A–C*). Western blot analysis of 4 dpf zebrafish larvae was performed to confirm that the increase in p53 protein expression observed in *mpi* morphants was reduced by co-injection of *p53* morpholino (*Figure 3D*). Notably, Mpi enzyme activity was not altered after Mpi knockdown with either co-injection of *p53* MO or with *mpi* MO in the p53 null mutants (*Figure 3E*), demonstrating the phenotypic rescue was not due to secondary increases in Mpi activity. Thus, Mpi loss induces p53, and this mediates the morphological abnormalities found in our zebrafish model.

Given the activation of p53 following Mpi-depletion, we asked whether the *mpi* morphant phenotype of reduced body size and small head, liver and intestine was a result of downstream consequences of p53 activation. Specifically, we assessed apoptotic cell death, using acridine orange (AO) staining and cleaved Caspase-3 immunofluorescence staining. *mpi* MO embryos stained positive throughout the embryo, having significantly increased positive foci compared to std MO controls, for both acridine orange (AO: N = 4, p=0.002) and cleaved Caspase 3 (Casp3: N = 3, p=0.016) when quantified in the tail (*Figure 3—figure supplement 1A–C*). Furthermore, cell death in *mpi* morphants was dependent on p53 since concurrent knockdown of *tp53* through co-injection of the *p53* MO significantly prevented the induction of apoptosis (*Figure 3B* and *Figure 3—figure supplement 1D*).

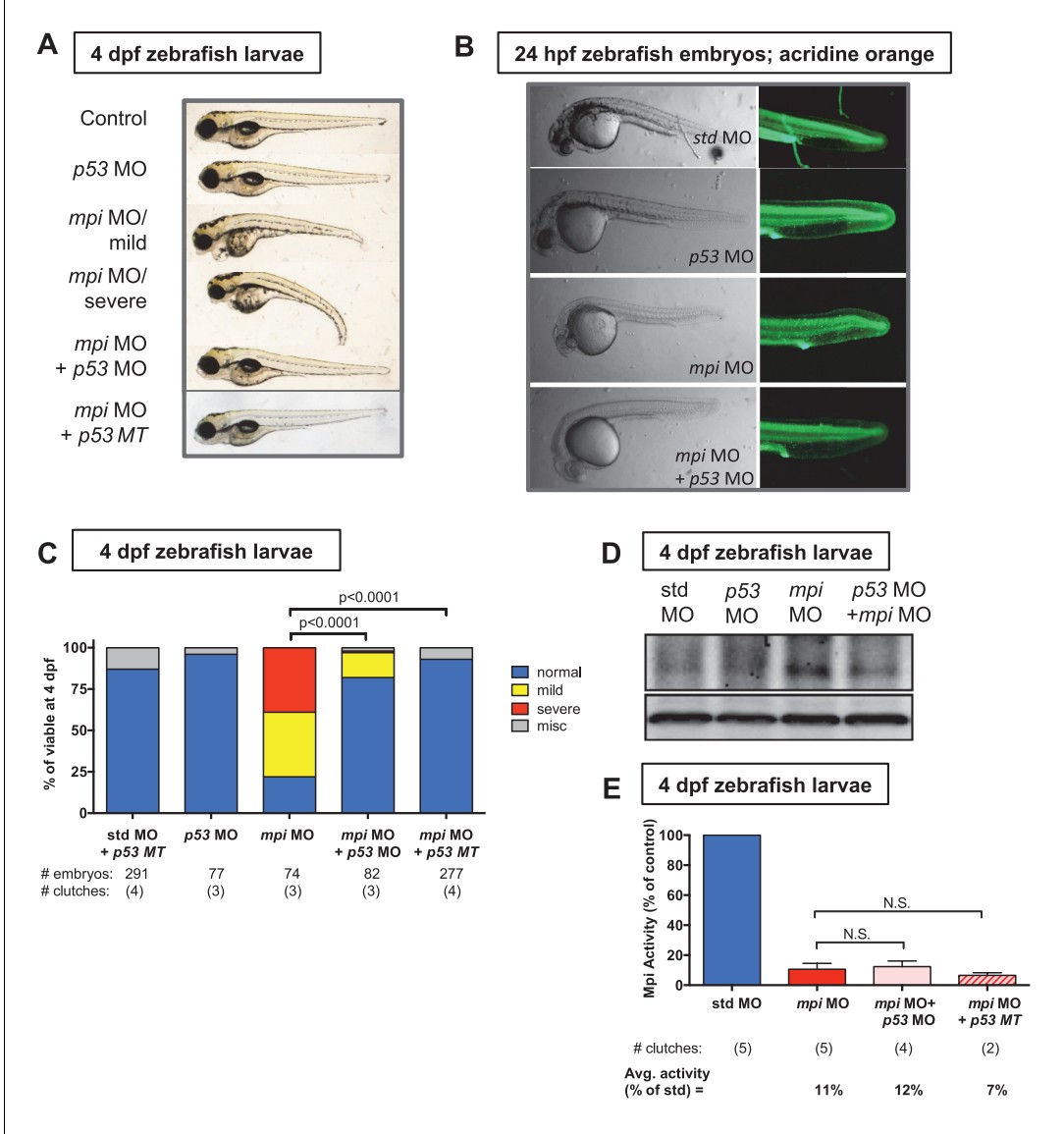

**Figure 3.** p53 is downstream of Mpi. (**A**) Representative phenotypes of 4 dpf zebrafish larvae following injections of std MO, *mpi* MO and *p53* MO. *mpi* morphants are characterized by a small head, microphthalmia, pericardial edema, jaw defects and reduced liver size. Examples of normal, mild and severe phenotypes are shown. Co-injection of *mpi* MO and *p53* MO, and injection of *mpi* MO into *p53* MT larvae showed phenotypic rescue of *mpi* knockdown. (**B**) The cell death phenotype in *mpi* morphants, visualized by acridine orange staining, was reversed by co-injection of *mpi* and *p53* morpholino. (**C**) Quantification of scoring of the phenotypes at 4 dpf, showing that 82% of *mpi* morphants injected with *p53* MO, and 93% of *p53* MT injected with *mpi* MO, were rescued to normal as compared with 22% normal in *mpi* MO larvae. p-values based on two-tailed Fisher's exact test. (**D**) Western blot analysis showing that increase in p53 protein expression in *mpi* morphants was reduced by co-injection with *p53* morpholino. Representative image of four separate clutches. (**E**) Mpi activity was not affected in either *p53* MO co-injected embryos or with *mpi* MO in the *p53* MT. p-Value based on two-tailed paired Student's t-test and Bonferroni correction was applied with alpha = 0.025; N.S. represents p-value>0.05.
The following figure supplements are available for figure 3:

**Figure supplement 1.** Cell death from loss of Mpi is rescued with p53 depletion.
**Figure supplement 2.** Tm-induced hypoglycosylation does not depend on p53.

To test whether this induction of p53-dependent apoptosis was specific to MPI loss or a response to global disruption in N-glycosylation, we used tunicamycin (Tm), a well established and widely used inhibitor of N-glycosylation, which we have used extensively in zebrafish (*Vacaru et al., 2014*; *Howarth et al., 2013*; *Cinaroglu et al., 2011*), to test whether phenotypes were similar to that of Mpi depletion. Importantly, Tm treatment yielded very different phenotypes compared to *mpi* MO embryos, indicating different pathological function in vivo (*Figure 3—figure supplement 2*). The efficacy of Tm in blocking glycosylation was confirmed using a transgenic zebrafish line that express human transferrin (*Tg(fabp10:hTf;cmlc2:EGFP)*), a clinical marker for hypoglycosylation in humans. Tm-treated embryos from this line demonstrated hypoglycosylation of human transferrin, as seen in zebrafish livers at 5 dpf (*Figure 3—figure supplement 2A*). In addition, Tm elicited gross pheno-types in embryos that were distinct from those observed in *mpi* MO; acridine orange staining in the Tm-treated embryos was notably absent (*Figure 3—figure supplement 2B*). Furthermore, disruption of N-glycosylation did not appear to be dependent on p53 as there was persistence of phenotypes in Tm-treated embryos in a *p53* MT background (*Figure 3—figure supplement 2C*). Together, these findings suggest that phenotypes following Mpi depletion are not a result of generalized disruption in N-glycosylation, and are specific to Mpi depletion. While this does not rule out that some of the Mpi-associated phenotypes could be caused by a modest defect in N-glycosylation, this instead highlights a new, alternative function of MPI in metabolism and p53 activation.

## Loss of MPI induces p53 in mammalian embryonic and cancer cells

The novel finding that p53 mediates the response to low Mpi levels in zebrafish embryos prompted us to examine if p53 is similarly induced following MPI depletion in mammalian cells. We utilized optimized Dicer-substrate interfering RNAs (dsiRNA) targeting mouse *Mpi* (siMpi) to decrease MPI function in primary mouse embryonic fibroblasts (MEF) (*Figure 4A–B*). We assessed the efficiency and the consequence of *Mpi* knockdown in MEFs using two independent siMpi sequences (siMpi-2 and siMpi-3; *Figure 4—figure supplement 1A–B*, and *Supplementary file 1*). In the subsequent experiments, using the most effective construct (siMpi-2; herein referred to as siMpi), the average residual MPI enzymatic activity was 43% (N = 8, p=0.002; *Figure 4A*) and MPI protein level decreased to 35% (N = 3, p=0.048; *Figure 4B*) compared to cells transfected with non-targeting control dsiRNA (NC). MPI-depleted MEFs demonstrated higher expression of p53, at both the tran-scriptional level, as measured by qPCR (1.6-fold increase, N = 8, p=0.007; *Figure 4C* and *Figure 4—figure supplement 1C*), and at the protein level, measured by western blot densitometry (2.1-fold increase, N = 5, p=0.02; *Figure 4D*), and is consistent with the results following Mpi depletion in zebrafish embryos (*Figure 2A,C and D*, and *Figure 2—figure supplement 2B–C*).

Many important pathways utilized during embryonic development are abnormally re-activated in cancer cells (*Dang, 2012*; *Fiske and Vander Heiden, 2012*; *Vander Heiden et al., 2009*). We asked whether MPI loss in cancer cells would also lead to increased p53 levels. To examine this, we depleted MPI using small-hairpin RNA (shRNA) in Hepa1-6 mouse liver cancer cells, and dsiRNA-mediated knockdown in SJSA human osteosarcoma cancer cells (*Figure 4—figure supplement 1D–E*). In both cases, we detected a substantial increase in p53 protein levels (*Figure 4E*). We further found an inverse relationship between *MPI* mRNA expression and p53 target gene expression in human tumor samples. Analysis of published microarray datasets for 118 primary hepatocellular car-cinomas (*Hoshida et al., 2009*) revealed that induction of p53, indicated by a p53 target gene signa-ture (*Kannan et al., 2001*), was negatively correlated with *MPI* expression levels (p=0.016; *Figure 4F*). These findings imply that loss of MPI induces p53 in cancer cells as well as in embryos. This is consistent with a recent finding that *Mpi* knockdown in glioblastoma cells confers radiosensi-tivity (*Cazet et al., 2014*), a phenotype that is directly associated with p53 induction.

## MPI loss suppresses the Warburg effect

MPI is well described for its function in generating precursors used for protein N-glycosylation. How-ever, as described above, our unbiased RNA-seq results from Mpi-deficient zebrafish embryos did not show any substantial alteration of genes involved in N-glycosylation (*Figure 2—figure supple-ment 1B*). In contrast, we found that *mpi* morphant embryos had decreased expression of many genes involved in glycolysis (*Figure 5A* and *Figure 5—source data 1*). Notably, *glucose-6-phos-phate isomerase (gpib)* and *phosphofructokinase, muscle (pfkmb, and pfkma)*, which encode

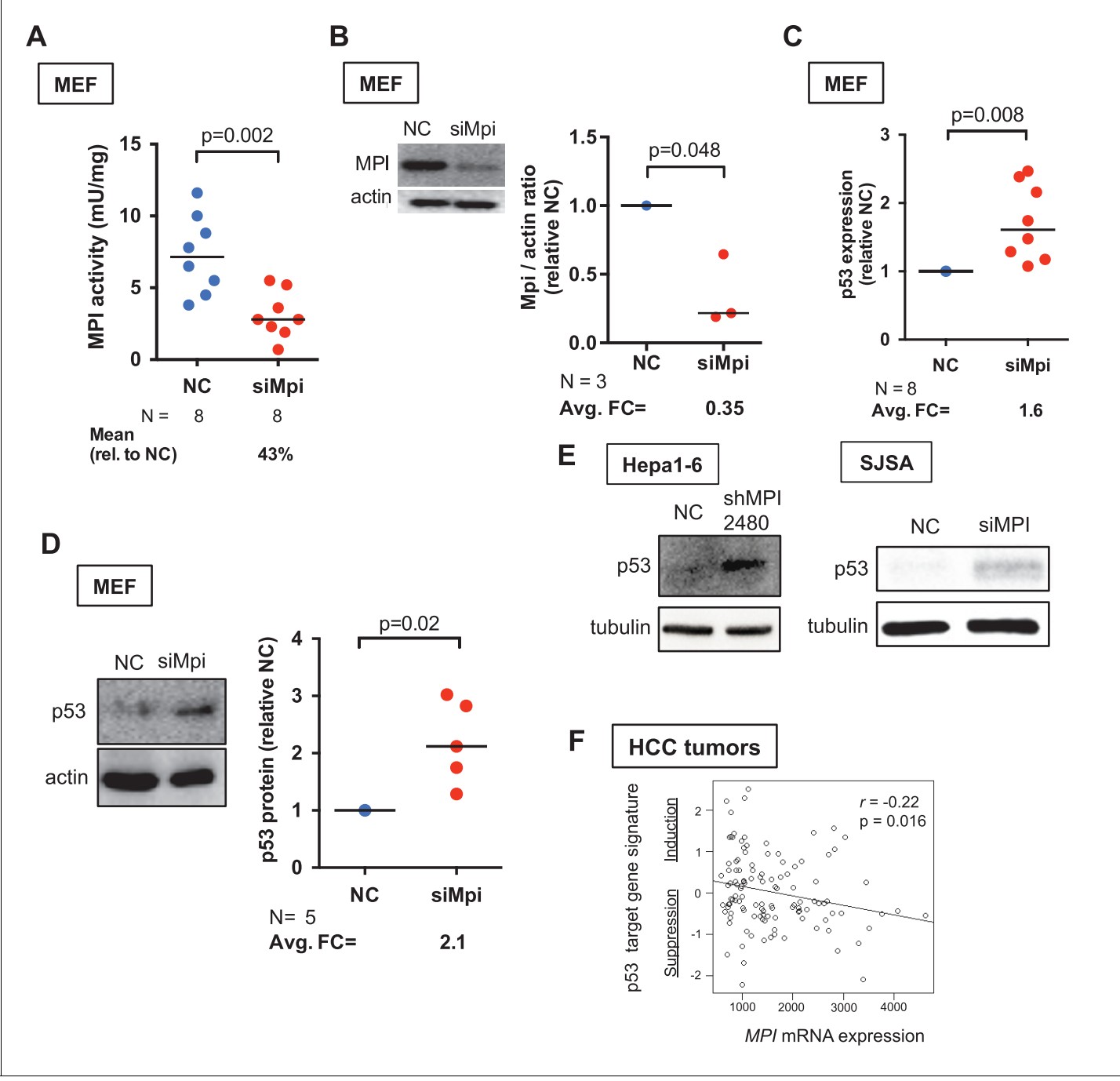

**Figure 4.** MPI knockdown induced p53 in embryonic and cancer cells. (**A**) Efficiency of MPI knockdown shown by reduction in MPI activity in MEFs transfected with siMpi compared to negative control siRNA (NC). Absolute activity units (mU/mg protein) are shown. (**B**) Western blot analysis showing a significant decrease in MPI protein levels in siMpi MEFs compared to NC cells. Western blots were quantified using densitometry analysis (ImageJ) and normalized to actin. (**C**) Measurement of *p53* mRNA in *Mpi* knockdown MEFs by qPCR demonstrated significant increase in *p53* expression in siMpi compared to NC. The Ct values were normalized relative to the expression of *Rps28* gene. Relative fold change was calculated as delta Ct in siMpi compared to NC. (**D**) Western blot analysis showing an increase in p53 protein levels in siMpi MEFs compared to NC. Western blots were quantified using densitometry analysis (ImageJ), and the results were normalized using actin. Two-tailed paired Student's t-test was applied in panels A-D. (**E**) Western blot analysis showing an increase in p53 protein levels in cancer cells (Hepa1-6 and SJSA) following MPI knockdown by shRNA or by dsiRNA, respectively, compared to NC. Images are representative of five independent (Hepa1-6) and two independent (SJSA) blots. (**F**) Transcriptome profiles of 118 human hepatocellular carcinoma (HCC) tumors showing that a p53 target gene signature induction is inversely correlated with *MPI* expression level with statistical significance. Pearson correlation test was applied.

*Figure 4 continued on next page*

*Figure 4 continued*

The following figure supplement is available for figure 4:

**Figure supplement 1.** MPI can be efficiently depleted in mammalian cell lines.

enzymes for the first two (including rate-limiting) steps of glycolysis, were the most downregulated genes in *mpi* morphants, and were each expressed over 4-fold lower than controls (log2 fold change −2.6,–2.4 and −2.2, respectively; *Figure 5—source data 1*). We next assessed whether MPI depletion similarly downregulates glycolytic genes in mammalian embryonic cells. Using qPCR, we followed the expression of *Pfkm* and *lactate dehydrogenase* (*Ldha*) in primary MEFs. LDH is responsible for the final step in glycolysis and has been shown to be critical in the Warburg effect and tumor growth (*Jiang et al., 2016*; *Arora et al., 2015*; *Zhou et al., 2010*). We found that *Pfkm* was significantly decreased by 34% (N = 5, p=0.011) and *Ldha* expression was reduced by 45% (N = 5, p=0.029) in MPI-depleted MEFs (*Figure 5B*).

Given that transcriptional modulation of glycolytic genes has been shown to signal depressed glycolytic flux (*Diedrich et al., 2016*; *Chen et al., 2014*), we hypothesized that the transcriptional decrease we discovered following MPI depletion would functionally translate to decreased glycolytic activity. To test the glycolytic response to MPI loss, we measured glucose uptake and lactate secretion, both of which are important indicators of the Warburg effect in embryos and in cancer (*Faubert et al., 2013*; *Fiske and Vander Heiden, 2012*; *Zhang et al., 2011*). In primary MEFs following MPI depletion, we found that glucose uptake was decreased to 53% of controls (N = 8, p<0.0001, *Figure 5C*), and secreted lactate levels were decreased to 49% of controls (N = 5, p<0.0001; *Figure 5D*). We next assessed suppression of Warburg metabolism in *mpi*$^{mss7}$ mutant and *mpi* MO embryos. In pooled embryos from incrossed *mpi*$^{mss7/+}$ adults (residual Mpi activity of 57%; *Figure 2—figure supplement 3F*), lactate was decreased to 63% compared to control WT incrosses (N = 4 clutches, p=0.033; *Figure 5E*). Similarly, *mpi* morphants had reduced lactate levels to 74% of controls (N = 6 clutches, p=0.008; *Figure 5F*).

Collectively, these data suggest that glycolysis is suppressed following MPI depletion, and we predicted that this should lead to reduced cell proliferation and viability. To examine this, we counted the number of cells in culture following MPI depletion. We found that in primary MEFs and SJSA cells, knockdown of MPI led to a 26% decrease in cell number in MEFs (N = 2, p=0.016; *Figure 5—figure supplement 1A*), and a 41% decrease in cell number in SJSA cells (*Figure 5—figure supplement 1B*). We independently found a similar decrease in cell proliferation and viability in MEFs using an MTT cell proliferation assay, which revealed MPI-deficient MEFs to have 69% signal to that of controls (N = 5, p=0.002; *Figure 5—figure supplement 1C*). This is consistent with the lethal phenotype observed following depletion of Mpi in zebrafish mutant and morphant embryos (*Figure 2—figure supplement 3D–E*, and [*Chu et al., 2013*]).

To further investigate the shared metabolic pathways between embryonic and cancer cells, we depleted MPI from human colon cancer cells (HCT116) in which Warburg metabolism has been shown to be essential for tumor cell proliferation (*Hussain et al., 2016*; *Wanka et al., 2012a*). The transfection efficiency for HCT116 cells was lower than that for MEFs, resulting in 62% residual MPI activity following siRNA transfection (N = 6, p=0.001; *Figure 5—figure supplement 2A*). This moderate decrease in MPI activity still resulted in reduced expression of key glycolytic genes, including *PFKM* (31% decrease, N = 4, p=0.04), *PKM1* (21% decrease, N = 4, p=0.047), *PKM2* (40% decrease, N = 4, p=0.004) and *LDHA* (30% decrease, N = 4, p=0.026; *Figure 5—figure supplement 2B*). Lactate levels were decreased as well (23% reduction, N = 6, p=0.03; *Figure 5—figure supplement 2C*). MPI knockdown in HCT116 cells also led to depletion of ATP to 69% of controls (N = 8, p=0.001; *Figure 5—figure supplement 2D*). These results indicate that MPI depletion also suppresses glycolysis in cancer cells.

p53 has been shown to be a negative regulator of glycolysis (*Bensaad et al., 2006*; *Hu et al., 2010*), and we sought to determine whether the induction of p53 is responsible for the glycolytic suppression seen following Mpi knockdown. We compared RNA-seq results with Mpi depletion in WT versus *p53* MT embryos and found that *p53* mutation suppressed the transcriptional changes in genes encoding glycolytic enzymes found in *mpi* morphants (*Figure 5A*, right column). To determine

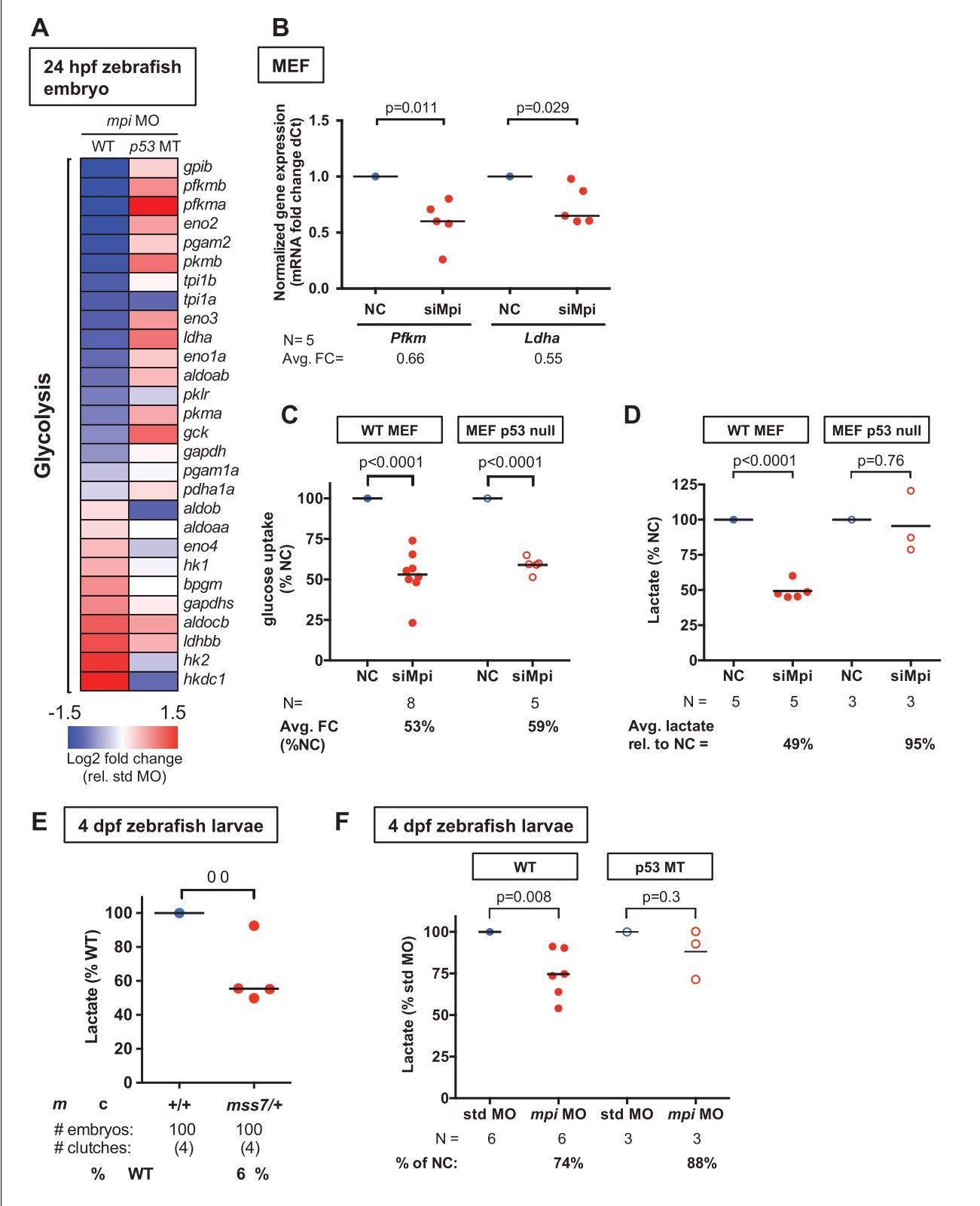

**Figure 5.** MPI loss suppresses the Warburg effect in embryonic cells. (**A**) Heat map based on RNA-seq data analysis showing downregulation of glycolytic genes in *mpi* morphants at 24 hpf. Standardized fold change (FC) in *mpi* morphants in WT and *p53* MT was calculated relative to standard control morpholino (std MO)-injected embryos. The color scheme represents gene expression changes in a log2 scale, in the range of −1.5 (blue, decreased) to 1.5 (red, increased). (**B**) qPCR analysis showed decrease of *Pfk* and *Ldha* mRNA levels in siMpi MEFs compared to NC. The Ct values

*Figure 5 continued on next page*

*Figure 5 continued*
were normalized relative to the expression of *Rps28* gene. Relative fold change was calculated as delta Ct in siMpi compared to NC. (**C**) 2-Deoxyglucose uptake measurement demonstrated a decrease in siMpi MEFs compared to the NC. Normalized relative % change is shown. (**D**) Measurement of lactate levels in MEFs following siMpi-mediated depletion of *Mpi* was assessed by lactate assay and showed a significant reduction of lactate in siMpi cells. Results were normalized to MTT assay. Relative amounts of lactate are shown (% of NC). (**E**) Measurement of lactate levels in 4 dpf zebrafish embryos from *mpi^mss7/+* MT incrosses showed a decrease in lactate production compared to the offspring of the *mpi^+/+* incrosses. Results were normalized to number of embryos. (**F**) Measurement of lactate levels in 4 dpf zebrafish embryos injected with std MO or *mpi* MO showed significant decrease in lactate production in *mpi* MO-injected embryos compared to std MO injected WT embryos, but no decrease when injected in to p53 MT embryos. Results were normalized to number of embryos. Two-tailed Student's t-test was applied in panels B-F.
The following source data and figure supplements are available for figure 5:

**Source data 1.** Changes in expression of the genes involved in glycolysis.
**Figure supplement 1.** Loss of MPI leads to decrease cell viability in embryonic and cancer cells.
**Figure supplement 2.** Loss of MPI leads to inhibition of glycolysis in HCT116 cancer cells.

whether p53 mutation or loss could reverse the functional repression of glycolysis observed with MPI depletion, we examined glucose uptake and lactate levels in zebrafish embryos, MEFs, and HCT116 cells, all with well-characterized p53 loss-of-function (LoF) counterparts. p53 LoF suppressed the reduction in lactate levels in all three systems (*Figure 5D and F*, and *Figure 5—figure supplement 2C*), a result consistent with restoration of glycolytic activity. However, p53 LoF had a negligible effect on preventing decreased glucose uptake following MPI depletion in p53 null MEFs (41% decrease, N = 5, p<0.0001) when compared to WT MEFs (*Figure 5C*). Together, our findings in zebrafish embryos and mammalian embryonic and cancer cell lines point to a previously unreported role for MPI in maintaining glycolysis and Warburg metabolism needed for cell survival.

## Loss of MPI activates p53 through the hexosamine biosynthetic pathway

MPI effectively links the N-glycosylation pathway with glycolysis and the hexosamine biosynthetic pathway (HBP), as all three pathways begin with Fru6P as substrate (*Figure 1*). MPI has primarily been studied for its role in converting Fru6P to Man6P for protein N-glycosylation (*de la Fuente et al., 1986*; *Fujita et al., 2008*; *Sharma et al., 2011*; *Cline et al., 2012*). We therefore predicted that MPI depletion would result in increased Fru6P levels. Using a sensitive fluorometric coupled enzyme assay, we measured intracellular Fru6P in MEFs following *Mpi* knockdown and found a significant accumulation of Fru6P (N = 4, p=0.027; *Figure 6A*). Given the suppression of glycolysis we observed with MPI depletion, we hypothesized that the accumulation of Fru6P found after MPI depletion would be metabolized into pathways that also use Fru6P as substrate, other than N-glycosylation and glycolysis. We focused on the HBP, which can be triggered under metabolic stress conditions (*Guillaumond et al., 2013*; *Buse, 2006*; *Chaveroux et al., 2016*), and hypothesized that having the excess of Fru6P caused by Mpi deficiency would increase the supply of Fru6P substrate entering the HBP (*Figure 1*). To test this, we blocked HBP using 6-diazo-5-oxo-L-norleucine (DON), a known chemical inhibitor of Glutamine—fructose-6-phosphate transaminase 1 (GFPT1, or GFAT1), the first step of the HBP (*Kim et al., 2009*; *Darley-Usmar et al., 2012*; *Sage et al., 2010*; *Marshall et al., 1991*). Embryos were treated with 20 µM DON immediately following either *mpi* or std MO injection. DON did not elicit a phenotype in std MO injected embryos and did not alter Mpi enzymatic activity levels in either controls or *mpi* morphants (N = 3; *Figure 6B–C and D*, respectively). However, DON treatment remarkably rescued the morphological phenotype of *mpi* morphants at 4 dpf (from 8.8% to 86.4% normal relative to untreated std MO, p<0.0001; *Figure 6B–C*). Most strikingly, p53 levels returned to near baseline by blocking HBP with DON in *mpi* morphants (*Figure 6E*).

DON is a glutamine analog and known inhibitor of GFPT1. However, DON also inhibits other enzymes that utilize glutamine as a substrate (*Cervantes-Madrid et al., 2015*). Therefore, to confirm our results were specific to HBP attenuation, we examined the effects of Mpi loss on Gfpt1 more

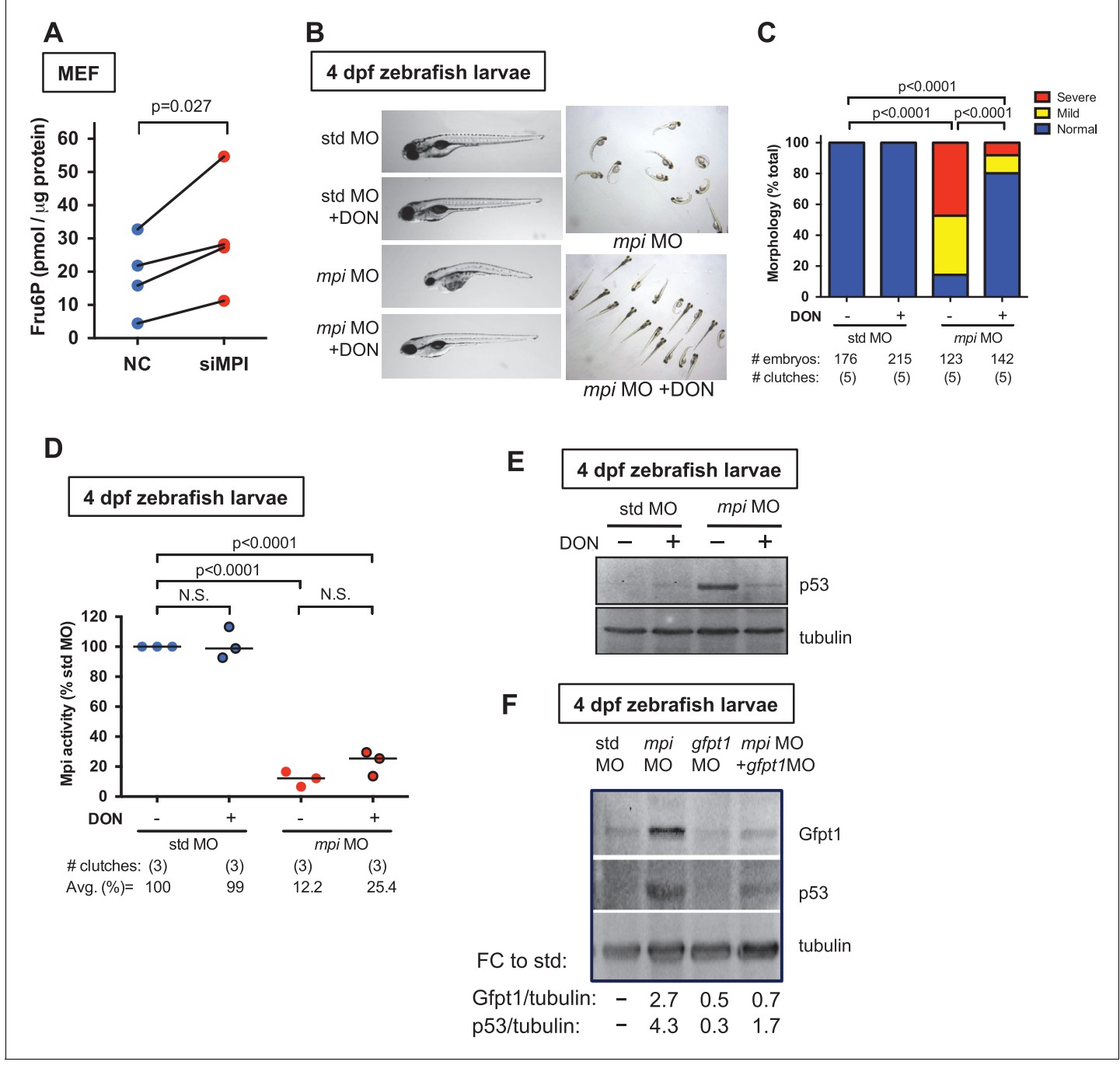

**Figure 6.** Loss of MPI stabilizes p53 through activation of the hexosamine biosynthetic pathway. (**A**) Measurement of Fru6P levels showed accumulation of Fru6P in siMPI MEFs compared to NC control. Values shown are normalized to total protein. p-Value based on paired ratio two-tailed Student's t-test. (**B**) Representative phenotypes of 4 dpf zebrafish larvae following injections of std MO or *mpi* MO and treated with 20 μM of HBP inhibitor (DON) are shown. Enlarged (left) and group (right) images are shown. (**C**) Distribution of the morphological phenotypes was quantified in 4 dpf *mpi* morphants and std MO control injected embryos treated with or without DON. Morphological rescue was observed, as an increase in embryos with normal morphology in the presence of the inhibitor, compared to *mpi* morphants. Two-tailed Fisher's exact test with the Bonferroni correction for multiple comparisons, with alpha = 0.017. (**D**) Mpi activity assay of 4 dpf zebrafish larvae injected with std MO or *mpi* MO and treated with 20 μM of DON showed that Mpi activity was not significantly affected by DON treatment. Two-tailed Student's t-test was applied with Bonferroni correction for multiple comparisons, with alpha = 0.0125. (**E**) Representative western blot for p53 protein in 4 dpf zebrafish larvae injected with std MO or *mpi* MO and treated with 20 μM DON showed a decreased p53 expression in *mpi* morphants treated with DON, compared to non-treated. DON treatment did not induce increase in p53 in std MO. Representative western blot for three independent clutches. (**F**) Western blot showing levels of Gfpt1 and p53 in

*Figure 6 continued on next page*

eLIFE Research article
Cancer Biology | Cell Biology

*Figure 6 continued*

4 dpf zebrafish embryos injected with either std, *mpi*, *gfpt1*, or *mpi* MO + *gfpt1* MO. Quantified relative to tubulin loading control, and compared to std MO. Representative western blot for four independent clutches.

specifically. Consistent with higher HBP activity, Mpi-depleted zebrafish embryos produced increased Gfpt1 protein levels (*Figure 6F*). More important, similar to DON treatment, knockdown of Gfpt1 with morpholino (*gfpt1* MO)(*Senderek et al., 2011*) dampened the p53 activation seen in *mpi* morphants (*Figure 6F*), suggesting the likely mechanism of p53 stabilization is through HBP function.

The induction of HBP in response to Mpi-depletion prompted us to investigate the mechanism by which Mpi depletion activates p53. UDP-GlcNAc, a product of HBP, serves as the donor for O-linked $\beta$-N-acetylglucosamine addition to proteins (O-GlcNAcylation; *Figure 1*). The addition of O-GlcNAc to p53 is a stabilizing post-translational modification that prevents p53 degradation (*Yang et al., 2006*). We hypothesized that MPI loss enhances HBP activity, which should increase protein O-GlcNAcylation, including p53 leading to its stability. To test this, we first examined total protein O-GlcNAcylation following MPI loss in zebrafish and found increased total O-GlcNAc staining in Mpi-depleted embryos at both 24 hpf and 4 dpf (*Figure 7A–B*), an effect that was counteracted by inhibition of Gfpt1 with DON (*Figure 7A*). These results were confirmed by directly targeting O-GlcNAc transferase (Ogt), the enzyme that catalyzes the transfer of GlcNAc to protein, with a specific morpholino against *ogt* mRNA (*ogt* MO)(*Webster et al., 2009*). Co-injection of *ogt* MO and *mpi* MO led to a clear reversal of the increased amounts of O-GlcNAcylated proteins following *mpi* MO alone (*Figure 7B*). Furthermore, consistent with activation of the HBP and increased Gfpt1 protein (*Figure 6F*), Ogt protein levels also increased following Mpi depletion (*Figure 7C*, lane 3 vs. lane 1). Together, these results suggest an overall increase in HBP activity following Mpi depletion.

As shown above, targeting Gfpt1 following Mpi depletion reverses p53 stabilization, leading to reduction in p53 levels (*Figure 6E–F*). Importantly, we also found reversal of p53 stabilization following Ogt targeting. Co-injection of *ogt* morpholino with *mpi* MO reduced p53 protein levels, compared to *mpi* MO alone, a result observed at both 24 hpf and 4 dpf (*Figure 7B–C*). We confirmed this result by using OSMI-1, a small molecule, cell permeable inhibitor of OGT that has been previously validated in multiple cell lines (*Ortiz-Meoz et al., 2015*). Mpi-depleted zebrafish embryos were treated with 50 µM OSMI-1 immediately following *mpi* MO injection. Similar to *ogt* MO, treatment with OSMI-1 was effective in mitigating p53 activation in *mpi* morphants, (*Figure 7C*). Collectively, these results suggest that Mpi loss activates p53 through HBP activity and increased O-GlcNAcylation.

Given that MPI loss increases total O-GlcNAc and p53 levels and that inhibiting HBP and O-GlcNAcylation corresponded with reduced p53 levels, we hypothesized that the mechanism of p53 stabilization was through p53 O-GlcNAcylation. To test this and extend our findings to human cancer cells, we depleted MPI in human osteosarcoma SJSA cells. Immunoflorescence staining for total O-GlcNAc levels confirmed our zebrafish findings and demonstrated an increase in total O-GlcNAcylation in SJSA cells following MPI depletion (*Figure 7D*). To address whether O-GlcNAcylated p53 was induced with MPI loss, we adapted a proximity ligation assay (PLA), which has been previously used to detect phosphorylation modifications using two specific antibodies (*Jarvius et al., 2007*). Here, we used PLA to visualize the O-GlcNAcylated form of p53 by using an anti-p53 antibody and an anti-O-GlcNAc antibody in SJSA cells. Following MPI knockdown, we found an increase in the level of O-GlcNAcylated p53 (*Figure 7E*). Additionally, the signal of O-GlcNAcylated p53 was markedly diminished when co-treated with DON to block HBP and O-GlcNAcylation (*Figure 7E–F*). Nuclear signal was not detected in control siRNA tranfected cells. Based on this data, we conclude that across vertebrate species, MPI loss stabilizes p53 by increasing Fru6P flux into HBP to promote p53 O-GlcNAcylation revealing an entirely new function for a classic metabolic enzyme.

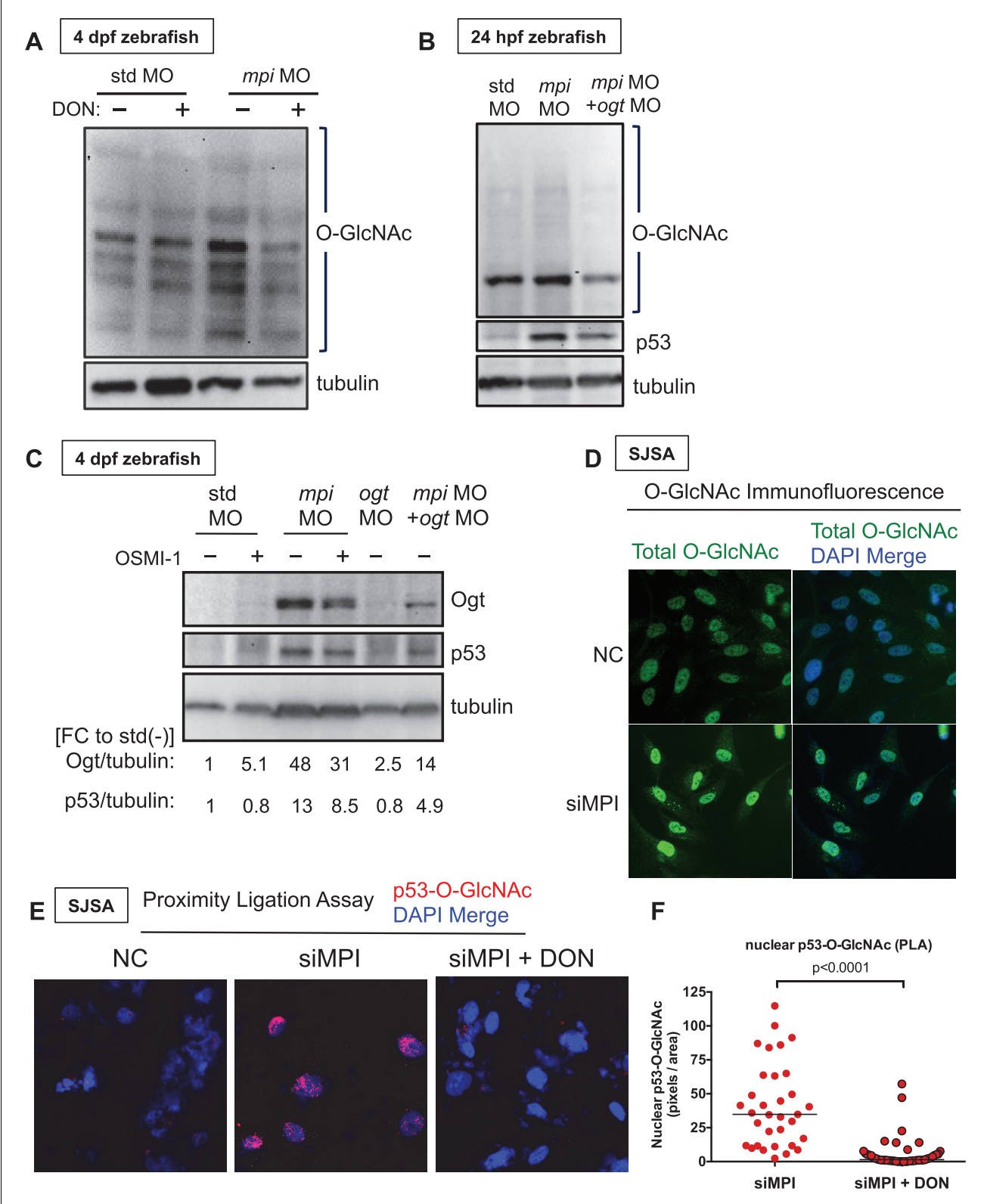

**Figure 7.** MPI depletion increases p53 O-GlcNAcylation. (**A–B**) Western blots of O-GlcNAc levels in std and *mpi* MO-injected zebrafish with either DMSO, 20 uM DON treatment, or *ogt* MO co-injection. Tubulin as loading control. (**C**) Western blot of Ogt and p53 levels in std, *mpi* and *ogt* MO with and without 50 uM OSMI-1 treatment. Tubulin used as loading control. Quantification of pixels performed using ImageJ is shown below blots. (**D**) Images of conventional immunofluorescence staining in SJSA cells following MPI knockdown showed that in siMPI cells there was a increase in total

*Figure 7 continued on next page*

*Figure 7 continued*

protein. (**E–F**) PLA showing increased O-GlcNAcylation of p53 with MPI knockdown that is diminished with DON treatment as compared to NC. 20x magnification was used from imaging using Zeiss inverted microscope Axiovert 1A. Image shown is representative of two independent experiments. Quantification using ImageJ software.

## Discussion

Embryonic and cancer cells rely on robust glucose metabolism to generate precursors for biosynthesis of lipids, nucleotides and amino acids and sustain rapid cell division. Due to this vital role in cell survival, regulation of glycolysis and Warburg metabolism requires careful orchestration by survival factors, metabolic enzymes, and metabolic intermediates themselves, via rate-limiting enzymatic steps, post-translational modifications and transcriptional changes (*Bensaad et al., 2006*; *Elstrom et al., 2004*; *Kruiswijk et al., 2015*; *Ozcan et al., 2010*). The central findings of this study reveal a previously unappreciated role for MPI in HBP and p53 regulation, which is conserved across species, and shared between embryonic and cancer cells. We find that that MPI loss: (1) suppresses glycolysis and (2) induces p53 by increased flux through HBP and increased O-GlcNAcylation. We believe this novel role for MPI in p53 regulation is distinct from, but not necessarily at the expense of, its traditional and well-studied role in N-linked glycosylation. These data demonstrating a novel function of MPI is strongly supported by the fact that the clinical presentation of MPI-CDG is so distinct from every other type of congenital disorder of glycosylation, as well as other MPI studies that cast doubt on the magnitude of impact that MPI has on generalized N-glycosylation (*DeRossi et al., 2006*; *Fujita et al., 2008*), although this may be partially attributable to varied experimental conditions and assay sensitivity. Here, we broaden our understanding of the relationship between this classic metabolic enzyme, MPI, and one of the most well studied tumor suppressor proteins, p53, to maintain glycolysis and sustain cell survival.

The functional consequences of MPI loss are striking: glycolysis is blocked and cells die. In a range of systems - zebrafish embryos, primary mouse embryonic fibroblasts, mouse and human cancer cell lines, and primary human liver tumors - loss of MPI induced p53, revealing a new mechanism of p53 regulation. This provides insight into the regulation of the Warburg effect and uncovers a previously unappreciated role for MPI as positive regulator of glycolysis, and surprisingly, as an important and underappreciated mediator of Fru6P, as MPI activity is required to prevent accumulation of Fru6P, a central glycolytic substrate. This accumulation of Fru6P may be due to primary effects of depletion of MPI enzymatic activity, but also may have contributions from regulation by p53, which has also been shown to play a role in the complex metabolic coordination and distribution of Fru6P between glycolysis, HBP, N-glycosylation, and the Pentose Phosphate Pathway (*Cheung and Vousden, 2010*; *Gottlieb, 2011*; *Kruiswijk et al., 2015*; *Lee et al., 2014*). As an example, p53 induces TIGAR, and regulates Fru6P metabolism to enhance PPP at the expense of glycolysis, in order to reduce ROS (*Wanka et al., 2012b*; *Bensaad et al., 2006*). We found that MPI depletion resulted in Fru6P accumulation, but downregulation of glycolysis as shown by decreased glucose uptake and decreased lactate in zebrafish, mouse, and human samples. Although p53 is a known suppressor of glycolysis, MPI knockdown on a p53-deficient background restored lactate to levels similar to control but did not improve glucose uptake. This is consistent with previous data showing that loss of p53 increases baseline lactate levels (*Gutierrez et al., 2010*), presumably removing suppression of glycolysis by p53. The persistent decrease in glucose uptake was not entirely surprising, as others have reported, both in vitro and in mice-bearing tumor xenografts, no difference in Fluoro-2-deoxyglucose uptake between HCT116 cells with wild–type p53 versus the HCT116 p53 null cells (*Wang et al., 2007*), suggesting that p53 is not solely responsible for suppression of glucose uptake, and that other factors likely contribute. Additionally, N-glycosylation of GLUT1 has been shown to be important for its protein stability and function; treatment of human leukemic cell lines with tunicamycin, an inhibitor of N-glycosylation, decreased 2-DG uptake by 40–50%, with a 2–2.5-fold decrease in GLUT1 affinity for glucose (*Asano et al., 1993*, *1991*; *Ahmed and Berridge, 1999*). Our data suggest that MPI loss downregulates glycolytic gene expression and lactate in a p53-dependent manner in zebrafish, mouse, and human cells, but the suppression of glucose uptake was not reversible with p53 loss. Given these findings, we cannot conclude that the suppression of glycolysis with MPI depletion is

wholly dependent on p53. The interplay between MPI and p53 with regard to regulation of glycolysis is likely complex and highlights the importance of understanding these complex relationships between pathways responsible for energy metabolism, namely in the setting of rapid cell proliferation in embryogenesis and cancer.

The HBP is usually a minor metabolic pathway yet it plays an important role in cancer (*Lefebvre et al., 2010*; *Jones et al., 2014*; *Jóźwiak et al., 2014*; *Wellen et al., 2010*). It generates UDP-GlcNAc which serves as a metabolic sensor and a precursor for posttranslational modification and nuclear trafficking of oncogenic transcription factors, including PI3K, PFK, and p53, all of which have been shown to be important regulators of glycolysis (*Jóźwiak et al., 2014*). p53 has been shown to be stabilized in response to stress via O-GlcNAcylation (*Ozcan et al., 2010*; *Yang et al., 2006*), and this O-GlcNAc modification has been shown to be important to cancer cell growth (*de Queiroz et al., 2016*). Here, we found that loss of MPI leads to accumulation of Fru6P and activation of the HBP pathway, resulting in increased total O-GlcNAcylation, and specifically O-GlcNAcylation of p53. Inhibiting different steps of the HBP pathway (GFPT1 or OGT) either through genetic or chemical perturbation (DON or OSMI-1 inhibition, respectively) prevented p53 stabilization associated with MPI depletion. Interestingly, MPI-CDG patients are treated with oral mannose supplementation, which corrects the majority, but not all, of their symptoms (*de Lonlay and Seta, 2009*, *Niehues et al., 1998*; *Mention et al., 2008*). In addition to mannose bypassing MPI deficiency to supplement Man6P for N-glycosylation, it is possible that clinical improvement following mannose supplements could also be through a HBP-dependent mechanism. Mannose has been reported to reduce cellular levels of UDP-GlcNAc (*Jokela et al., 2008*), and so mannose supplementation may act to counteract the enhancement of HBP, and lower protein O-GlcNAcylation. Further investigation into the effects of mannose supplementation on MPI loss, O-GlcNAcylation, and p53 activation would be a topic for further study, with potential implications in therapeutic intervention.

While the complex pathways involved in metabolism and cell proliferation are still unfolding and are likely cell- and context-dependent, there is new focus on metabolic enzymes and their metabolites as central oncogenic players (*Ward and Thompson, 2012*). Our study places MPI among these important regulators and advances our understanding of metabolic programs that control the Warburg effect and cell survival both during embryonic development and in cancer. In the past decade, anticancer drug discoveries have targeted glucose metabolism through PI3K and PFK inhibitors, suppressors of lactate production, glutamine metabolism, and DNA replication, many of which are currently in clinical trials (*Wise and Thompson, 2010*; *Zhou et al., 2013*; *Doherty and Cleveland, 2013*). Our data suggest the exciting possibility that MPI could be a potent anticancer drug target, as depletion of MPI results in widespread dampening of glycolysis, halting cell growth and promoting death.

## Materials and methods

### Zebrafish maintenance and embryo injection

Adult zebrafish were maintained on a 14:10 hr light:dark cycle at 28°C. Wild-type (WT; AB, Tab 14) and *tp53* mutant (*Berghmans et al., 2005*) fish (courtesy of S. Sidi) were used. Fertilized eggs collected following natural spawning were cultured at 28°C in fish water (0.6 g/l Crystal Sea Marinemix; Marine Enterprises International, Baltimore, MD) containing methylene blue (0.002 g/l). The Mount Sinai School of Medicine Institutional Animal Care and Use Committee approved all protocols. Morpholinos targeting the ATG of the *mpi* transcript (5′-GAGGAAACACACTTTCACTTCCGCCAT-3′), targeting the ATG of the *p53* transcript (5′-GCGCCATTGCTTTGCAAGAATTG-3′), targeting the ATG of the *gfpt1* transcript (5′-TCAGATACGCAAATATGCCACACAT-3′)(*Senderek et al., 2011*), targeting the ATG of the *ogt* transcript (5′-CCACGTTCCCCACCGAGCTTGCCAT-3′) (*Webster et al., 2009*), and a standard control morpholino (5′-CCTCTTACCTCAGTTACAATTTATA-3′) that does not target any known zebrafish transcript were obtained from Gene Tools (Philomath, OR). Needles were calibrated to inject 2 nL per embryo using a Narishige IM-300 microinjector; 0.1–4 ng of morpholino per embryo was used; 1 ng of *mpi* MO was identified as optimal injection amount. All injections were carried out in one- to four-cell stage embryos.

## Zebrafish transgenic lines

Transgenic line (*Tg(fabp10:hTf;cmlc2:EGFP)*) expressing human transferrin (hTf) was obtained by injecting the corresponding construct into the one-cell-stage zebrafish embryos. At 48 hpf, the embryos were screened for the expression of the transgene in the expected tissue, and then raised until adulthood to create the founders of the line. These were then out-crossed with TAB14 wild type (WT), and the positive progeny was raised to create the F1 generation. Throughout the study we used F2 and F3 generations of (*Tg(fabp10:hTf;cmlc2:EGFP)*).

## Acridine orange

24 hpf zebrafish were manually dechorionated and placed in 15 mL dishes containing acridine orange solution (80 µL concentrated Acridine Orange solution in 40 mL egg water (0.6 g/l Crystal Sea Marinemix; Marine Enterprises International, Baltimore, MD) containing methylene blue (0.002 g/l)). Dishes were wrapped in foil and kept in the dark for 15–30 min. Embryos then went through a series of five clean egg water washes, transferred each time via a pipet in a net well. Embryos were mounted on a depression slide with 2% methylcellulose and imaged under green fluorescent light.

## Cleaved Caspase-3 immunostaining

Adapted from *Sidi et al. (2008)*, dechorionated embryos at 24 hpf were then fixed in 4% PFA overnight at 4°C, and dehydrated in methanol at −20°C for at least 2 hr. Embryos were rehydrated three times, 5 min each, in PBST (1xPBS, 0.1% Tween-20), and permeabilized by treatment with PDT (PBST + 1% DMSO) supplemented with 0.3% Triton-X for 20 min. Embryos were treated with blocking solution (PDT with 10% heat inactivated FBS, Boeringer Manheim Blocking Solution) for 30 min before the addition of primary antibody (anti-pH3, 1:750, anti-activated-Casp-3, 1:500). Embryos were incubated in primary antibody overnight at 4°C, rinsed three times, 20 min each, in PDT, then reblocked for 30 min in blocking solution before the addition of AlexaFluor-conjugated antiR a555 Red fluorescent secondary antibody (1:500) for 1 hr. Embryos were rinsed for 5 min in PDT and imaged under red fluorescent light.

## MPI enzyme assay

MPI activity assay was performed according to our previously published protocol in zebrafish (*Chu et al., 2013*) and adopted for mammalian cells extract. Briefly, zebrafish larvae or cell lysates were homogenized and protein concentration was determined by the Bradford assay (*Bradford, 1976*). MPI activity was assessed in 15 µg of protein extract and residual MPI activity following MPI depletion was calculated relative to the control samples (WT strain for *mpi^mss7^*, std MO for *mpi* MO, or NC for siMPI cells).

## Mammalian cells culture conditions

Primary mouse embryonic cells (MEFs, MEFs p53 null; courtesy of S.Lowe), human colon cancer cell lines (HCT116 (RRID:CVCL_0291), and HCT116 p53 null (RRID:CVCL_HD97); courtesy of B. Vogelstein), human osteosarcoma cell line SJSA (RRID:CVCL_1697; courtesy of Joaquín M. Espinosa) and mouse liver cancer cell line (Hepa1-6; RRID:CVCL_0327). These cell lines are not on the list of the commonly misidentified cell lines as established by the International Cell Line Authentication Committee. Cell lines were checked for mycoplasma using Venor GeM Mycoplasma Detection Kit (Cat# MP0025; Sigma-Aldrich, St. Louis, MO). Cells were cultured in Dulbecco's Modified Eagle's Medium, high glucose, with phenol red (DMEM, Cellgro, Manassas, VA), supplemented with 10% heat-inactivated fetal bovine serum (FBS, Invitrogen, Carlsbad, CA) and Penicillin-Streptomycin (Cellgro, Manassas, VA). Cells were passaged 1 day prior to experiments, using 1X trypsin-EDTA (0.25%, Cellgro, Manassas, VA).

## dsiRNA-mediated silencing

Culture media was replaced to antibiotic free media 2 hr before transfection. Duplex siRNA (dsiRNA, negative control (NC), validated negative control that is not present in the human, mouse, or rat genomes) and MPI-targeted siRNA (siMPI)) were ordered from IDT. Sequences listed in *Supplementary file 1*. dsiRNA was reconstituted in water to 10 µM and 1 µM siMPI concentrations for NC and siMPI, respectively. Transfection was performed according to the Life Technologies

protocol for siRNA transfection, using Lipofectamine RNAiMAX Transfection Reagent (ThermoFisher Scientific, Waltham, MA) and Opti-MEM I Reduced Serum Medium (ThermoFisher Scientific, Waltham, MA). The final concentration of the siRNA in a well was 1 nM and 10 nM for NC and siMPI, respectively. Transfection was repeated 48 hr after the first transfection, and cells were collected for assays 24 hr following the second transfection.

## shRNA -mediated silencing

Lentivirus carrying shRNA constructs targeting mouse *Mpi* mRNA were purchased from Broad Institute, Cambridge, MA and Hepa1-6 cells were infected using the protocols optimized by TRC Broad Institute. Briefly, 1.0E + 05 cells per plate were infected using 1 mg/ml polybrene per well and 160 μl of 10 iU/ml of the lentivirus carrying targeting or control shRNA. Media with virus was replaced with fresh media 17 hr after infection, and cells were passaged 1:2 48 hr after transfection and starts drug resistance selection next to non-transfected control (puromycin 2 μg/ml) for 3–5 days (until the nontransfected control with the drug dies). Efficiency of MPI knockdown following shRNA-mediated silencing was assessed using MPI activity assay. shRNA constructs were purchased from the Broad Institute. Sequences can be found in *Supplementary file 1*.

## Western blotting

Protein lysates were prepared from either 24 hpf or 4 dpf zebrafish. In the case of 24 hpf zebrafish, 30 eggs were dechorionated manually using syringe needles, and deyolked in 1X PBS (cold), and centrifuged at 11,963 g for 7.5 min at 4°C. These deyolked 24 hpf larvae, and ten to twenty 4 dpf larvae, were homogenized in lysis buffer (20 mM Tris pH 7.5, 150 mM NaCl, 1% NP-40, 2 mM EDTA, 10% glycerol and protease inhibitors). In cases of both time points, all lysates were centrifuged and 1:5 vol of sample buffer was added to the supernatant to achieve 2% SDS, 5% 2-mercaptoethanol. Samples were run on a 10% polyacrylamide gel and western blotted. The membranes were blocked for 1 hr in 5% skim milk in TBS-T (TBS with 10% Tween 20 (vol/vol)). Membranes were probed with primary anti-p53 (*MacInnes et al., 2008*) (1:2, mouse hybridoma) and anti-zebrafish p53 (1:1,000, GTX128135, Genetex, Irvine, CA), anti-human p53 (1:200, rabbit sc-6243 (RRID:AB_653753), mouse sc-99 (RRID:AB_628086), Santa Cruz Biotechnology, Santa Cruz, CA), anti-O-GlcNAc [RL2] (1:1,000, ab2739, Abcam, Cambridge, MA; RRID:AB_303264), anti-OGT (1:1,000, SAB2108697, Sigma-Aldrich, St. Louis, MO), anti-GFPT1 (1:1,000; 14132–1, Proteintech, Rosemont, IL; RRID:AB_11146805), anti-tubulin (1:2000; Developmental Studies Hybridoma Bank, Iowa City, IA; RRID:AB_1157911), anti-actin (1:2000, A2228 Sigma-Aldrich, St. Louis, MO), or rabbit anti-MPI antibody (generated by Hudson Freeze laboratory [*Davis et al., 2002*]), at 4°C. All antibodies were prepared in 2% milk in 1X TBST except anti-O-GlcNAc (2% BSA in 1X TBST). Following primary antibody incubation, membranes were incubated in HRP-conjugated secondary antibody (anti-mouse or anti-rabbit; Promega, Madison, WI), and were visualized by chemiluminescence using ChemiDoc XRS Imaging System (BioRad, Hercules, CA). Quantification of band intensities was performed using ImageJ software (http://rsbweb.nih.gov/ij/download.html).

## RNA extraction and qPCR analysis

Five embryos at 24 hpf or 4 dpf, or cells from one well in a 6-well plate, were homogenized in 1 ml of TRIzol (Invitrogen, Carlsbad, CA) and purified, using chloroform extraction (1:6) and isopropanol precipitation (1:1). cDNA was prepared by polyA priming using qScript SuperMix (Quantabio, Beverly, MA). See Supplementary Table 4 for primer sequences. qRT-PCR analysis was performed in the Light Cycler 480 (Roche, Rotkreuz, Switzerland) using gene-specific primers (see Supplementary materials) and PerfeCTa SYBRGreen FastMix (Quantabio, Beverly, MA). Ct values from triplicate reactions were averaged and 2-Ct(target)/2-Ct(reference) was used to calculate expression, with *rpp0*, *Rps28* or *RPS18* used as references genes of zebrafish, mouse or human samples, respectively.

## RNA sequencing

Total RNA was isolated from 20 zebrafish embryos at 24 hpf for each experimental group (std MO in WT, *mpi* MO in WT, std MO in *p53* MT, *mpi* MO in *p53* MT) using an RNA isolation and purification kit (Ambion, Austin, TX). Sequencing was performed at the Genomics Core of the Icahn School of Medicine at Mount Sinai. RNA integrity was confirmed by Agilent 2200 Tapestation with the R6K

ScreenTape (Agilent, Santa Clara, CA). The sequencing library was prepared with the standard Tru-Seq RNA Sample Prep Kit v2 protocol (Illumina, San Diego, CA). Briefly mRNA was isolated and fragmented. cDNA was synthesized using random hexamers, end-repaired and ligated with appropriate adaptors for sequencing. The library then underwent size selection and purification using AMPure XP beads (Beckman Coulter, Brea, CA). The appropriate Illumina recommended 6 bp barcode bases are introduced at one end of the adaptors during PCR amplification step. The size and concentration of the RNA-seq libraries was measured by Bioanalyzer and before loading onto the sequencer. The mRNA libraries were sequenced on the Illumina HiSeq 2500 System with 100 nucleotide single-end reads, according to the standard manufacturer's protocol (Illumina, San Diego, CA).

## RNA-seq analysis

We used exact matches to map the reads to the zebrafish genome (Zv9) and estimate the coverage of each gene (*Aravin et al., 2007*; *Olson et al., 2008*). Briefly, the reads were split into three 32 bp pairs after trimming 2 nt at each end, and the parts were mapped to the genome using a suffix-array based approach (detailed in [*Gurtowski et al., 2010*]). The mappings were then converted to an expression level by using the median of coverage across the transcript as an estimate of gene expression. The expression values were quantile normalized, and ratios were calculated by comparing the mean of the samples from *mpi* morphants against the mean from controls. We also used the coverage to cluster the data and determine outliers, which were excluded from further analyses. For mRNA-Seq expression data, the peak in the density distribution of expression values was used to estimate the noise in the system. The values were regularized by adding the noise to each gene's expression level before the ratios were calculated. This ensures that genes with low expression do not contribute to the list of genes with large fold changes, so that the differentially expressed genes are significant. Fold change was calculated using equation *lg(FC)= lg ((stdMO + noise) / (mpiMO + noise))* and used to generate the heat map.

## Fructose 6-phosphate measurement

Fructose 6-phosphate was quantified in primary MEFs using a coupled enzyme assay to stoichiometrically convert Fru6P to glucose 6-phosphate, with a final fluorometric readout of the NADPH-dependent conversion of resazurin to resorufin, as previously described for measurement of other sugar phosphates (*Zhu et al., 2011*; *Sharma et al., 2014*). Sugar phosphates were extracted by lysing cell pellets in 125 µl 1M perchloric acid with sonication, centrifugation, and neutralization of the supernatant with 125 µl 2M potassium bicarbonate. Twenty microliters neutralized extract was measured per assay, in duplicate, using an enzyme cocktail consisting of PGI, G6PD and diaphorase, with the fluorometric substrate resazurin supplied, and NADP added. Fluorescence was measured with an excitation wavelength of 535 nm, and emission wavelength of 590 nm, using a SpectraMax M5 microplate reader. Samples without PGI were subtracted from the same samples with PGI added (Fru6P → Glc6P) to determine total Fru6P-based fluorescence in samples, and quantified by comparing to fluorescence measured using a standard curve with Fru6P as substrate. Total Fru6P per sample was normalized to total protein measured using BCA assay (Pierce, Waltham, MA) by re-suspending the protein pellet after perchloric acid extraction in 10 mM Tris-Cl, 2% SDS.

## TALEN mutant line generation

Transcription activator-like effector nucleases (TALENs) were designed to target exon 2 of the *mpi* gene, generating a double-stranded break, instigating error-prone non-homologous end joining. GoldyTALENs were designed using the MojoHand software (*Neff et al., 2013*) and assembled using a modified protocol (*Ma et al., 2013*). Sequences generated were as follows:

TAL1: 5'-CCTCTGTGCTGTGTG-3' 22-—36 (15)
Spacer: 5'-GTGCAGAACTACGCCTG-3' 37-—53 (17)
TAL2: 5'-GGGTAAAGCGGGTCTGG-3' 54-—70 (17)
Binding Strand (reverse complement): 5'-CCAGACCCGCTTTACCC-3'. 65 ng/µL of the TALEN mRNA was injected into one-cell embryos. F0 embryos were randomly pooled and genomic DNA was extracted from 2 to 5 dpf embryos in 100 mM NaOH, heated at 95°C for 15 min, and neutralized with 1M Tris-HCl. TALEN insertion was verified by PCR amplification for exon 2 (forward primer: 5'AGACATGGCGGAAGTGAAAG-3'; reverse primer: 5'TCTGCATAGGGTTTGCTGTG-3') of the *mpi*

gene followed by a BsgI restriction digest. Presence of the *mpi* mutation (mss7) was indicated by an uncut band at 500 bp, as visualized on a 2% agarose gel (*Figure 2—figure supplement 3B*), and confirmed by sequencing (data not shown). F0 embryos harboring *mpi* mutations were raised and outcrossed (after 3 months upon adult maturation) to WT (AB, Tab14) adult zebrafish to generate the F1 generation, and genotyped as described. Heterozygous F1 fish were incrossed to generate the F2 generation, comprising wild-type, heterozygous and homozygous fish. Because of homozygous embryonic lethality by 13 dpf, the *mss7* mutation was maintained as heterozygous, and heterozygous mutants were incrossed to generate homozygous embryos used for experiments. Where indicated in the text, offspring from *mpi$^{mss7/+}$* incrosses were were separated based on genotype for analysis, or pooled and compared to offspring from *mpi$^{+/+}$* siblings.

## Glucose uptake assay

2-Deoxyglucose uptake was estimated in a 96-well plate by an enzymatic NADPH amplifying system assay (ab136955, Abcam, Cambridge, UK). Briefly, 1500 MEFs were serum starved overnight. The cells were then incubated with 100 µl KRPH buffer containing 2% BSA for 40 min at 37°C. 10 µM 2-Deoxyglucose was added to the cells for 20 min to simulate glucose uptake in cells. 2-Dexyglucose was metabolized by cells into 2-Deoxyglucose-6-phosphate, which was then oxidized to generate NADPH. Measurements of NADPH levels were made using a recycling amplification reaction method, and glucose uptake was estimated based on 2-deoxyglucose uptake. Samples were measured at OD412 nm using a kinetic plate reader every 2 min. Results were corrected for protein content with a BCA protein assay kit.

## Lactate assay

For measuring lactate synthesis in cells we used Lactate assay kit (Trinity Biotech, Jamestown, NY) and modified manufacturer protocol. Briefly, we added 10 µl of conditioned media collected from the cells to 200 µl of lactate reagent in 96-well plate and incubated for 15 min at RT. Lactate standard curve (0, 20, 80, and 120 mg/dL) was used for quantification. Light absorbance was measured at 540 nm. The results of the lactate assay were normalized to total protein measured by Bradford assay.

For measuring lactate synthesis in zebrafish embryos, 4 dpf larvae were homogenized in 100 mM Tris, 4 mM EDTA buffer (20 µl/20 fish), heated at 95°C for 5 min and centrifuged at 12,000 x g at RT. Ten microliters of the supernatants were used for lactate assay as described above. The results of the lactate assay were normalized to embryo number.

## ATP quantification

ATP was quantified in HCT116 cells using ENLITEN ATP Assay System (Promega) according to the manufacturer protocol. Cells collected from 1 well in a 6-well plate were lysed in 100 µl of 6% perchloric acid and aqueous fraction was collected by centrifugation. The supernatant was neutralized and clarified by 30 µl of KCO3 and additional centrifugation. Of the supernatant, 50 µl was used for measurement of luminescence and quantified based on the standard curve. The absolute numbers were normalized using MTT assay.

## MTT cell proliferation assay

Mitochondrial activity of each cell line using the tetrazolium dye-based micro-titration assay to measure mitochondrial dehydrogenases activity as described elsewhere (*Boamah et al., 2007*). Briefly, MTT solution (5 mg/mL MTT powder dissolved in balanced salt solution without phenol red) was added to the cells in an amount equal to 10% of the culture medium volume and incubated at 37°C in 5% CO2 for 2 hr. The cells were centrifuged at 451xg for 5 min at RT, and pellets were resuspended in 0.04 N hydrochloric acid diluted in isopropanol. Samples were incubated for 5 min at RT and centrifuged at 22,000xg for 2 min. Supernatant absorbance was read at 550 nm (620 nm absorbance was subtracted for background). Data represent percent population of viable cells in each sample relative to the untreated sample.

## Hexosamine biosynthetic pathway inhibitor treatment

*mpi* morphant zebrafish embryos and standard control morpholino-injected embryos were placed in fish water containing either 20 µM 6-diazo-5-oxo-L-norleucine (DON; Sigma-Aldrich, St. Louis, MO) or 50 µM OSMI-1 (Sigma-Aldrich, St. Louis, MO) at 2 hpf and cultured at 28°C in fish water. Embryos were collected for gene and protein expression analysis at 24 hpf or 4 dpf and scored for morphological evaluation. SJSA cells were treated with 25 µM DON beginning 2 hr following siMPI transfection.

## Proximity ligation assay (PLA) and immunofluorescence imaging

PLA was used for detection of O-GlcNAcylation modification on human p53 protein using SJSA cells, primary antibodies specific for general O-GlcNAc and p53 (SCBT, Santa Cruz, CA) and Duo-link in situ Fluorescence kit (Sigma-Aldrich, St. Louis, MO). Cells were fixed using Paraformaldehyde (4% in PBS), permeabilized using Triton (0.1% in PBS) and incubated with primary antibody in PAXDG (PBS containing 1% BSA, 0.3% Triton X-100, 0.3% deoxycholate, and 5% bovine serum). The consecutive steps were according to the manufacturer protocol (Sigma-Aldrich, St. Louis, MO). Total O-GlcNAcylation was detected using fluorescent secondary antibody. PLA and IF was visualized using a Zeiss inverted microscope Axiovert 1A.

## Statistical analysis

All statistical analyses were performed using the statistical package built into Prism software (GraphPad Software, La Jolla, CA). Where multiple comparisons were performed for a single set of experiments, the alpha level for significance was adjusted using the Bonferroni correction; otherwise significance level was set at 0.05. Specific statistical tests that were performed for each set of experiments, as well as which alpha value was used, are indicated in the respective figure legends.

## Acknowledgements

Funded by National Institutes of Health [K08DK101340 to JC, 5R01DK080789 and 6R01AA018886 to KCS, R01DK99551 to HHF, T32DK007792 to CD], Gilead Sciences Research Scholars Program in Liver Disease to JC, The Rocket Fund to HHF, and the Mindich Child Health and Development Institute at Mount Sinai to JC. We thank Patrick Bradley, Matthew Nash, Chen Lu, and Louis Lin for technical support. RNA sequencing was performed at the Genomics Core Facility at the Icahn School of Medicine at Mount Sinai. TALENs were assembled by the Genetics and Model Systems Core (M McNulty, K Clark and SEkker) in the Mayo Clinic Center for Cell Signaling in Gastroenterology (NIDDK P30DK084567).

## Additional information

### Funding

| Funder | Grant reference number | Author |
| --- | --- | --- |
| National Institute of Diabetes and Digestive and Kidney Diseases | K08 DK101340 | Jaime Chu |
| The Mindich Child Health and Development Institute at Mount Sinai | | Jaime Chu |
| National Institute of Diabetes and Digestive and Kidney Diseases | R01DK080789 | Kirsten C Sadler |
| National Institute on Alcohol Abuse and Alcoholism | R01AA018886 | Kirsten C Sadler |
| National Institute of Diabetes and Digestive and Kidney Diseases | R01DK99551 | Hudson H Freeze |
| The Rocket Fund | | Hudson H Freeze |

| National Institute of Diabetes and Digestive and Kidney Diseases | T32DK007792 | Charles DeRossi |
| --- | --- | --- |

The funders had no role in study design, data collection and interpretation, or the decision to submit the work for publication.

## Author contributions
NS, Conceptualization, Data curation, Formal analysis, Investigation, Methodology, Writing—original draft, Writing—review and editing; CD, Conceptualization, Data curation, Formal analysis, Writing—original draft, Writing—review and editing; SN, Data curation, Formal analysis, Validation, Investigation, Writing—original draft, Writing—review and editing; RS, Data curation, Formal analysis, Methodology; LSK, Conceptualization, Data curation, Formal analysis; AP, APK, YHa, Data curation, Formal analysis; AV, Data curation, Validation; YHo, DKS, Data curation, Formal analysis, Supervision; EE, Conceptualization, Data curation, Formal analysis, Supervision, Investigation, Methodology; HHF, Conceptualization, Writing—original draft, Writing—review and editing; KCS, Conceptualization, Supervision, Funding acquisition, Writing—original draft, Writing—review and editing; JC, Conceptualization, Data curation, Formal analysis, Supervision, Funding acquisition, Investigation, Methodology, Writing—original draft, Project administration, Writing—review and editing

## Author ORCIDs
Charles DeRossi, http://orcid.org/0000-0001-5267-6553
Kirsten C Sadler, http://orcid.org/0000-0002-1100-4125
Jaime Chu, http://orcid.org/0000-0002-9291-8630

## Ethics
Animal experimentation: This study was performed in strict accordance with the recommendations in the Guide for the Care and Use of Laboratory Animals of the National Institutes of Health. All of the animals were handled according to approved institutional animal care and use committee (IACUC) protocols (#IACUC-2015-0050) of the Icahn School of Medicine at Mount Sinai.

# Additional files

## Supplementary files
• Supplementary file 1. Sequences of the qPCR primers and dsiRNA, shRNA duplexes.

## Major datasets
The following previously published dataset was used:

| Author(s) | Year | Dataset title | Dataset URL | Database, license, and accessibility information |
| --- | --- | --- | --- | --- |
| Hoshida Y, Nijman SM, Kobayashi M, Chan JA, Brunet JP, Chiang DY, Villanueva A, Newell P, Ikeda K, Hashimoto M, Watanabe G, Gabriel S, Friedman SL, Kumada H, Llovet JM, Golub TR | 2009 | Transcriptome profiles of surgically resected HCC tumor tissues | https://www.ncbi.nlm.nih.gov/geo/query/acc.cgi?acc=GSE10186 | Publicly available at the NCBI Gene Expression Omnibus (accession no: GSE10186) |

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
