## [Decision Letter]

Thank you for submitting your article "MPI depletion enhances O-GlcNAcylation of p53 and suppresses the Warburg effect" for consideration by *eLife*. Your article has been favorably evaluated by Ivan Dikic (Senior Editor) and three reviewers, one of whom, Ralph DeBerardinis (Reviewer #1), is a member of our Board of Reviewing Editors. The following individual involved in review of your submission has agreed to reveal their identity: Richard M White (Reviewer #2).

The reviewers have discussed the reviews with one another and the Reviewing Editor has assembled a list of essential additional work that the reviewers feel must be completed before a recommendation may be made concerning the publication of your work. Please consider the details below and return a letter with your responses, and action plan to complete this work and a timetable for the return of a revised manuscript. The Editor will then consult the reviewers and respond with a recommendation.

Summary:

This manuscript reports that silencing or mutating Mannose Phosphate Isomerase (MPI) perturbs zebrafish development by inducing a p53-dependent gene expression program and cell death. Strikingly, the developmental defects in zebrafish were completely reversed when both MPI and p53 were silenced together, implicating p53 as the key mediator of developmental aberrations downstream of MPI loss. Although MPI silencing appeared to have little effect on N-linked glycosylation, it led to an accumulation of its substrate, fructose-6-phosphate (F6P), which also supplies hexosamine biosynthesis and O-GlcNAcylation. The authors provide evidence for enhanced O-GlcNAcylation and resulting stabilization of p53 when MPI is silenced.

Essential revisions:

1) There needs to be better evidence supporting the claim that the MPI mutant does not alter N-linked glycosylation. The reported evidence is weak; the data from a key western blot in Figure 2—figure supplement 1 are not quantified, and this blot demonstrates appearance of a hypoglycosylated protein upon MPI silencing (this band is partially cropped out). The authors should carefully characterize and quantify several products of N-linked glycosylation.

2) Regarding rescue of MPI loss by silencing p53, why does the p53 morpholino induce p53 protein levels (Figure 3). This is a major concern because it raises the possibility that the phenotypes are related to increased rather than decreased p53 levels. This logic also applies to the genetic mutant, since induction of the mutant isoform would confound the interpretation. A much better experiment would be to generate a true null using p53 CRISPR; this is a rapid and standard approach in zebrafish.

3) The implications of the mannose rescue data are not clear. If O-GlcNacylation of p53 rather than loss of N-linked glycosylation is responsible for the phenotype of MPI deficiency, then why does mannose (a precursor for N-linked glycosylation) rescue the phenotype? If there is an alternative way of explaining this result, the authors need to point it out. It would also be useful to know whether mannose supplementation prevents the metabolic defects of MPI depletion and induction of p53.

4) A key point of the paper is that MPI deficiency stabilizes p53 via activation of hexosamine biosynthesis. A western blot is needed to demonstrate that total O-GlcNac levels are higher in MPI mutants, and OGT silencing should be used to ensure specificity of the O-GlcNAc signal.

5) The authors need to provide more evidence that MPI silencing is not simply inducing p53 expression, as suggested by Figure 3. The paper also needs a more rigorous assessment of the abundance of O-GlcNAcylated p53. This could include quantitation of the PLA assay as well as an orthogonal approach, such as immunoprecipitating p53 and blotting with an anti-O-GlcNAc antibody. Is p53 O-GlcNAcylation in MPI silenced cells reversed by inhibiting hexosamine biosynthesis?

6) The authors should test whether the effects of MPI deficiency on glucose metabolism/cell viability in mammalian cells requires p53, as it does in zebrafish.

7) DON is a glutamine analog that inhibits most enzymes that use glutamine as a substrate. The DON experiments need to be complemented by genetic experiments to reduce GFAT activity more specifically.

8) Key conclusions involving siRNAs, for example those in Figure 4, should be validated with two independent siRNAs rather than one.

---

## [Author Response]

*Essential revisions:*

*1) There needs to be better evidence supporting the claim that the MPI mutant does not alter N-linked glycosylation. The reported evidence is weak; the data from a key western blot in Figure 2—figure supplement 1 are not quantified, and this blot demonstrates appearance of a hypoglycosylated protein upon MPI silencing (this band is partially cropped out). The authors should carefully characterize and quantify several products of N-linked glycosylation.*

Thank you for this important point. MPI has been thus far known only as an enzyme important for N-linked glycosylation, as other cellular roles have not yet been reported. The important finding in our study is that MPI has a function other than that for synthesis of N-glycans, namely in regulation of p53 stability via O-GlcNAcylation, and glycolytic regulation, and that this is not necessarily at the expense of N-glycosylation.

The aim of this study is not to discredit previous work on MPI, but instead to show a new function for this protein. We believe the phenotypes that we observe are not driven by changes in N-glycosylation, but did not intend to claim that N-glycosylation is normal with MPI depletion. We realize that this is not clear in the present manuscript and have removed the western blot in Figure 2—figure supplement 1. Instead, we further clarified in the text that MPI deficiency in humans (MPI-CDG) presents differently than the 40+ other CDG types that affect N-glycosylation, and this suggests that loss of MPI may have other consequences than simply deficient N-linked glycosylation.

To provide full transparency and address the reviewers’ concerns, we include the uncropped western blot image that is presented in the original Figure 2—figure supplement 1 (Figure 8). We believe the bottom band is a non-specific signal as its size is smaller than the expected transferrin size, and is not altered with de-glycosylating PNGase treatment. As additional new data, we have repeated this experiment using a different glycoprotein (Gc), which, similarly, does not show altered mobility when Mpi is depleted as compared to Tunicamycin treatment, a well-established inhibitor of N-glycosylation (Figure 8). But we agree, these data do not prove that N-glycosylation is unchanged with Mpi depletion, and would not include this at the current time.

Author response image 1.**DOI:**
http://dx.doi.org/10.7554/eLife.22477.021

Instead, we have added new Figure 3—figure supplement 2, to address whether early disruption in N-glycosylation has similar phenotypes to our Mpi-depleted zebrafish embryos, and whether this is dependent on p53. To do this, we used tunicamycin, which we have previously used extensively in zebrafish. Tm treatment yields very different phenotypes when compared to mpi MO embryos: in Tm treated embryos, we confirm reduced glycosylation of human Transferrin using Tg(fabp10:hTF;cmlc2:EGFP) treated with Tm (new Figure 3—figure supplement 2). In Tm-treated embryos, we do not observe cell death with acridine orange staining as compared to mpi MO, and disruption of N-glycosylation does not appear to be dependent on p53 as we see persistence of phenotypes in Tm-treated embryos in a p53 MT background (new Figure 3—figure supplement 2). Together, these findings suggest that phenotypes following Mpi depletion are not a result of disrupted N-glycosylation. However, we agree that this does not rule out that modest defects in N-glycosylation that may contribute to the Mpi-associated phenotypes. Instead, this study aims to highlight a novel function of MPI in interrelated metabolic pathways, namely the hexosamine biosynthetic pathway and O-GlcNAcylation.

*2) Regarding rescue of MPI loss by silencing p53, why does the p53 morpholino induce p53 protein levels (Figure 3). This is a major concern because it raises the possibility that the phenotypes are related to increased rather than decreased p53 levels. This logic also applies to the genetic mutant, since induction of the mutant isoform would confound the interpretation. A much better experiment would be to generate a true null using p53 CRISPR; this is a rapid and standard approach in zebrafish.*

The band seen with injection of p53 morpholino is most likely secondary to background induction of p53 that is similar to that seen with injection of standard control MO. The p53 antibody used in this study was developed and first reported by MacInnes et al.PNAS, 2008. In that report, they also showed increased p53 protein expression in p53 MO-injected embryos as compared to uninjected controls, shown in MacInnes et al., PNAS 2008, Figure 1. In our paper, we use standard control morpholino injection to control for any toxicity from injection, and to account for variability between clutches, we have quantified 5 western blots to show that the levels of p53 can vary from clutch-to-clutch, but we see no significant difference in p53 between standard control MO- and p53 MO-injected embryos (Figure 9). However, there is a consistent and marked increase in p53 levels following Mpi-depletion, and this is reversed when p53 is depleted (Figure 9, Figure 3, and new Figure 3).

Author response image 2.**DOI:**
http://dx.doi.org/10.7554/eLife.22477.022

To address the second issue raised, namely that induction of mutant p53 may occur in the p53 M214K MT used and would be better addressed by generating a p53 null CRISPR mutant: While this is a valid goal, we were not be able to complete this in a reasonable time frame for resubmission and feel that it would not change the central point of our study, as we already use four complementary methods to examine loss of p53 – p53 morpholino, a well-established p53 mutant in zebrafish, *p53^-/-^* null MEFs, and *p53^-/-^* null HCT116 cells.

*3) The implications of the mannose rescue data are not clear. If O-GlcNacylation of p53 rather than loss of N-linked glycosylation is responsible for the phenotype of MPI deficiency, then why does mannose (a precursor for N-linked glycosylation) rescue the phenotype? If there is an alternative way of explaining this result, the authors need to point it out. It would also be useful to know whether mannose supplementation prevents the metabolic defects of MPI depletion and induction of p53.*

Thank you for this point. As suggested, the mechanisms of mannose rescue, even in MPI-CDG patients, are not clear, and may occur through multiple pathways. While we believe this to be a very interesting and important investigation that should be addressed in the future, it is not a central point for the present study. As such, we have removed the mannose rescue data (old Figure 3—figure supplement 1) from the revised manuscript. Instead, as suggested, we have further elaborated on this in the Discussion section:

“Interestingly, MPI-CDG patients are treated with oral mannose supplementation, which corrects the majority, but not all, of their symptoms (de Lonlay and Seta, 2009, Niehues et al., 1998, Mention et al., 2008). […] Further investigation into the effects of mannose supplementation on MPI loss, O-GlcNAcylation, and p53 activation would be a topic for further study, with potential applications in therapeutic intervention.”

*4) A key point of the paper is that MPI deficiency stabilizes p53 via activation of hexosamine biosynthesis. A western blot is needed to demonstrate that total O-GlcNac levels are higher in MPI mutants, and OGT silencing should be used to ensure specificity of the O-GlcNAc signal.*

Done. Please see new Figure 7.

This was an excellent suggestion, and we have spent the majority of our resubmission period focused on points 4, 5, and 7, to strengthen our investigation into the mechanism of how MPI regulates metabolic pathways and p53 through O-GlcNAcylation. We systematically approached this novel pathway of MPI regulation of p53, from HBP and its rate limiting enzyme (GFAT), to total O-GlcNAc, and more specifically to OGT, the key enzyme in O-GlcNAcylation.

In new Figure 6, we have addressed reviewer comment #7, and further strengthened our finding that MPI loss activates HBP, with new data to show that Mpi depletion leads to increased Gfat by western blot, and that inhibition of HBP through either chemical inhibition using DON or genetic knockdown of its rate limiting enzyme, GFAT, can both dampen Mpi-induced p53 activation (new Figure 6).

Next, as the reviewers suggested in point #4 and #5, we expanded our analysis on the end product of the HBP, protein O-GlcNAcylation, and also depleted the key enzyme in this step, OGT. To address reviewers’ point #4, in new Figure 7, we demonstrate that loss of Mpi in zebrafish results in an increase in total O-GlcNAc and OGT levels (new Figure 7). This new data complements our immunofluoresence data to show increased total O-GlcNAcylation in mammalian cells (Figure 7). As suggested in point #4, we ensured specificity of the O-GlcNAc signal by inhibiting GFAT and Ogt in Mpi knockdown, which both decreased O-GlcNAc levels (new Figure 7).

As suggested in point #5, this revised manuscript presents additional compelling evidence that the mechanism of p53 induction by Mpi loss is through O-GlcNAcylation of p53 (new Figure 7, and 7F). In this revised manuscript, we show 4 different methods of inhibiting the HBP and O-GlcNAcylation, which all block MPI-induced p53. This is shown through 2 genetic approaches (silencing Gfpt1 and Ogt), and 2 chemical approaches with DON and OSMI-1, a small molecule inhibitor of Ogt (new Figure 7). We find that both total O-GlcNAc and p53 levels in Mpi-depleted zebrafish are diminished in all cases. Lastly, new data demonstrates that the O-GlcNAcylated p53 is reversed by inhibiting the HBP using DON, as assessed by PLA (new Figure 7) and quantified in new Figure 7 as requested.

Of note, we first approached this by designing and injecting 5 different CRISPRs to ogt in zebrafish, of which only 3 worked partially to induce a mutation in F0s. However, there was variable efficacy across 3 different experiments. Therefore, the ability to use CRISPR to generate a mutation in ogt was not feasible for the short term. We also attempted to further complement our findings with IP of p53 and O-GlcNAc in Mpi-depleted zebrafish, but had technical difficulties and were unable to optimize the assay in time for submission.

The reviewers’ suggestions were instrumental in our efforts to provide compelling evidence that MPI is a critical metabolic enzyme, not only for N-glycosylation, but in our uncovering of new functions for MPI in the regulation of HBP, O-GlcNAcylation, and specifically as a novel regulator of p53.

*5) The authors need to provide more evidence that MPI silencing is not simply inducing p53 expression, as suggested by Figure 3. The paper also needs a more rigorous assessment of the abundance of O-GlcNAcylated p53. This could include quantitation of the PLA assay as well as an orthogonal approach, such as immunoprecipitating p53 and blotting with an anti-O-GlcNAc antibody.*

Done. Please see new Figure 6, Figure 7, and the response to point #4.

*Is p53 O-GlcNAcylation in MPI silenced cells reversed by inhibiting hexosamine biosynthesis?*

Done. Please see new Figure 7, and the response to point #4.

*6) The authors should test whether the effects of MPI deficiency on glucose metabolism/cell viability in mammalian cells requires p53, as it does in zebrafish.*

Done. Please see new Figure 5 (RNA-seq of glycolytic genes in mpi MO in *p53^-/-^* MT embryos), 5C (Glucose uptake in MEF^+/-^ p53), 5D (lactate in MEF^+/-^ p53), 5F (lactate in zebrafish^+/-^ p53), and new Figure 5—figure supplement 2 (lactate in HCT116^+/-^ p53).

Thank you for this suggestion. p53 has been shown to be a negative regulator of glycolysis (Bensaad et al., 2006, Hu et al., 2010) and we aimed to determine whether the induction of p53 is responsible for the glycolytic suppression seen following Mpi knockdown. We compared RNA-seq results with Mpi depletion in WT versus p53 MT embryos and found that p53 mutation suppressed the transcriptional changes in genes encoding glycolytic enzymes found in mpi morphants (new Figure 5, right column). To determine whether p53 mutation or loss could reverse the functional repression of glycolysis observed with MPI depletion, we examined glucose uptake and lactate levels in zebrafish embryos, MEFs, and HCT116 cells, all with well-established p53 loss of function (LoF) counterparts. p53 LoF suppressed the reduction in lactate levels in all three systems (new Figure 5, and Figure 5—figure supplement 2), a result consistent with restoration of glycolytic activity. However, p53 LoF had a negligible effect on preventing decreased glucose uptake following MPI depletion in p53 null MEFs (41% decrease, N=5, p<0.0001) when compared to wt MEFs (new Figure 5).

The discrepancy seen with p53-dependent restoration of lactate levels, but not glucose uptake, is not entirely surprising. This is consistent with previous reports showing that loss of p53 increases baseline lactate levels (Gutierrez et al., 2010), presumably removing suppression of glycolysis by p53. The persistent decrease in glucose uptake despite p53 knockdown has been previously published in vitro and in mice bearing xenografted tumors, which showed no difference in Fluoro-2-deoxyglucose uptake between the HCT116 cells with wild‑type p53 versus the HCT116 *p53^-/-^* null cells (Wang et al., 2007), suggesting that p53 is not solely responsible for suppression of the glucose uptake, and that other pathways likely contribute. Additionally, N-glycosylation of GLUT1 has been shown to be important for its protein stability and function; treatment of human leukemic cell lines with tunicamycin, an inhibitor of N-glycosylation, decreased 2-DG uptake by 40-50%, with a 2-2.5-fold decrease in GLUT1 affinity for glucose (Asano et al., 1993, Asano et al., 1991, Ahmed and Berridge, 1999). Our data suggest that MPI loss downregulates glycolytic gene expression and lactate in a p53-dependent manner in zebrafish, mouse, and human cells, but the suppression of glucose uptake was not reversible with p53 loss. Given these findings, we cannot conclude that the suppression of glycolysis with MPI depletion is wholly dependent on p53. The interplay of MPI and p53 on glycolysis is likely complex, and highlights the importance of understanding these complex relationships between pathways responsible for energy metabolism, namely in the setting of rapid cell proliferation in embryogenesis and cancer. This is an interesting area for future study, and this has been added to the revised Results and Discussion sections.

*7) DON is a glutamine analog that inhibits most enzymes that use glutamine as a substrate. The DON experiments need to be complemented by genetic experiments to reduce GFAT activity more specifically.*

Done. Please see new Figure 6, new Figure 7, and the response to point #4.

*8) Key conclusions involving siRNAs, for example those in Figure 4, should be validated with two independent siRNAs rather than one.*

Done. Please see Figure 10 for summary of knockdown strategies used across 3 different vertebrate species, using both embryonic and cancer cells. We have knocked down MPI using 2 independent siRNAs in MEFs (Figure 4—figure supplement 1), and added new data for 2 independent shRNA for Hepa1-6 cells (new Figure 4—figure supplement 1). We have also confirmed these findings using siRNA in human SJSA cells, siRNA in HCT116 cells, and zebrafish mpi MT and mpi MO. In all cases, we see consistent depletion of MPI activity and induction of p53, which presents exciting and compelling data to uncover a new function of MPI that is conserved among vertebrates, and critical in both embryonic and cancer cells.

Author response image 3.**DOI:**
http://dx.doi.org/10.7554/eLife.22477.023